# Computationally efficient aerodynamic modeling of swept wind turbine blades using coupled near wake and vortex cylinder models

Ang Li<sup>1</sup>, Mac Gaunaa<sup>1</sup>, and Georg Raimund Pirrung<sup>1</sup>

<sup>1</sup>Department of Wind and Energy Systems, Technical University of Denmark, Frederiksborgvej 399, DK-4000 Roskilde, Denmark

Correspondence: Ang Li (angl@dtu.dk)

**Abstract.** This study introduces a computationally efficient engineering aerodynamic model specifically designed for load calculations of swept wind turbine blades, overcoming limitations in existing models. The proposed method couples a near wake trailed vortex model with a novel far wake vortex cylinder model. In this coupled model, the near wake, defined as the first quarter revolution of the blade's own trailed wake, is modeled using non-expanding helical vortices. Together with the influence of the curved bound vortex, the sweep effects are effectively captured. This comprehensive approach accounts for the influence of a finite number of blades, eliminating the need for Prandtl's empirical tip-loss correction used in conventional blade element momentum (BEM) methods. The far wake, representing the remaining trailed wake, is modeled using concentric vortex cylinders originating downstream of the rotor plane, replacing the conventional momentum-based approach. The near and far wake contributions are coupled together to obtain the total induction. In this study, a detailed analysis identifies limitations in the original coupling method, leading to two proposed modifications that enhance numerical stability and accuracy. Comparisons with higher-fidelity free-wake lifting line (LL) and Reynolds-averaged Navier-Stokes (RANS) simulations demonstrate the load prediction improvements, particularly for forward swept blades. The model achieves comparable accuracy with significantly reduced computational efforts, making it an ideal tool for design optimization and repetitive aeroelastic simulations of swept wind turbine blades. While developed and validated under steady-state conditions, the formulation readily supports extensions to unsteady aerodynamics using methodologies analogous to unsteady BEM approaches. The model can also be adapted in future work for generalized blade geometries combining sweep and prebend.

# 1 Introduction

Technological advancements in wind turbine design and manufacturing have led to substantial increases in the size and flexibility of modern horizontal-axis wind turbine (HAWT) blades, especially compared to those from the 1980s. For instance, a state-of-the-art offshore wind turbine installed in 2024 reached a rotor diameter of 260 m and a rated power of 18 MW. Innovative blade designs, such as backward swept blades for passive load alleviation through geometric bend-twist coupling (Liebst, 1986; Zuteck, 2002; Larwood and Zutek, 2006; Manolas et al., 2018) and aerodynamically or aeroelastically optimized curved blade tips involving sweep (Barlas et al., 2021; Madsen et al., 2022), are gaining popularity. Swept blades are also adopted in small wind turbine designs. For example, Eocycle integrates swept blade designs into its EOX M-series turbines

in distributed wind applications (Eocycle Technologies Inc., 2021a, b). Fritz et al. (2024a) conducted a wind tunnel experiment on backward swept blades using particle image velocimetry (PIV) methods. Additive manufacturing is also being explored for blade tips, enabling design integration and improved aerodynamics while reducing manufacturing constraints (Houchens et al., 2024). While these innovative designs offer potential aerodynamic and aeroelastic benefits, they also introduce increased modeling complexity, which exposes inherent limitations in existing aerodynamic models and poses significant challenges for load calculations and design optimization.

Higher-fidelity methods, such as free-wake lifting line (LL) solvers and Reynolds-averaged Navier–Stokes (RANS) simulations, provide relatively accurate load predictions but are computationally intensive, limiting their use to specific load cases or comparison studies. Consequently, the blade element momentum (BEM) method remains the primary tool for low-fidelity aerodynamic modeling of HAWTs, valued for its simplicity and minimal computational requirements, making it well-suited for design optimization and aero-servo-elastic simulations. However, BEM implicitly assumes a planar rotor with straight blades (Li et al., 2022b, 2025b, 2024). As a result, the BEM method cannot accurately predict the influence of wake geometry resulting from curved blade geometries or blade coning on aerodynamic loads, as shown in recent analytical and numerical studies (Li et al., 2022b, d, a; Barlas et al., 2022; Horcas et al., 2023; Li et al., 2025b, 2024; Zahle et al., 2024).

To mitigate some of BEM's limitations, Madsen and Rasmussen (2004) introduced a coupled near and far wake model that integrates a lifting line approach for the near wake with a momentum-based far wake model, building on the foundational work by Beddoes (1987). The near wake, defined as the first quarter revolution of the blade's own trailed vortex, is modeled using non-expanding helical vortex filaments, with an indicial function approach capturing its dynamic behavior. The remaining trailed wake, termed the far wake, is modeled based on momentum theory. The near and far wake models are coupled using a coupling factor to obtain the total induction (Andersen et al., 2010; Pirrung et al., 2016). The far wake inductions are determined by scaling down the BEM thrust coefficient using this factor during the induction calculation. Subsequent research extended this coupled model framework to swept blades by explicitly modeling the influence of curved bound vortices and accounting for the in-plane shift in the starting position of trailed vortices (Li et al., 2018, 2020, 2022d). Regarding accuracy in unsteady aerodynamic simulations, previous studies have shown that for a straight blade, the NW-MT approach can closely reproduce the aerodynamic response predicted by free-wake lifting line models, as demonstrated for pitch steps and prescribed vibration cases (Pirrung et al., 2017a). This represents a significant improvement over the BEM method.

Despite these advancements, the coupled model shows limitations when applied to forward swept blades, primarily due to limitations in the coupling factor and potentially the far wake model, both of which require further improvements (Li et al., 2022d). To address these challenges, this study introduces a novel far wake vortex cylinder model that replaces the existing momentum-based approach, providing a more physically consistent representation of the far wake. This study focuses on steady-state conditions (assuming uniform inflow, fixed rotor speed and no elastic deformations) to isolate sweep-induced effects. However, the framework is readily extendable to unsteady conditions using sub-models, such as dynamic inflow and dynamic stall models, similar to those applied in unsteady BEM methods (Madsen et al., 2020). In addition, this study refines the coupling factor through two modifications based on a modified Newton–Raphson approach, enabling automatic adjustments across different swept blade configurations, with improved numerical stability. Furthermore, the proposed model has the

60 potential to be further adapted to more generalized curved blades featuring combined sweep and prebend, which has significant implications for future wind turbine design optimization and aeroelastic simulations.

The remainder of this paper is structured as follows: Section 2 briefly reviews the inherent limitations of the BEM method in modeling swept blades. Section 3 outlines the existing coupled near and far wake model, which is extended for modeling swept blades. In Sect. 4, we develop the new far wake vortex cylinder model to enhance the physical representation of the far wake. Section 5 presents three modified coupled models with different fidelities that improve upon the existing coupled model for swept blades. In Sect. 6, we investigate and improve the coupling factor. Section 7 describes the simulation setups of higher-fidelity LL and RANS CFD solvers and also blade configurations used for comparison. Section 8 presents a comparative analysis of results from different models, evaluating the performance of the new models. Finally, Section 9 summarizes the contributions and findings of the study and suggests directions for future research.

## 2 Blade element momentum method

The blade element momentum (BEM) method is a fundamental engineering aerodynamic model that integrates blade element theory (BET) with momentum theory (MT). Due to its simplicity and minimal computational effort, BEM is widely used in aerodynamic load calculations of rotors. In the BEM method, the rotor plane is divided into multiple concentric annular sections, each assumed to be radially independent. Within each annulus, the aerodynamic forces acting on the 2-D airfoil sections (blade elements) of all blades are balanced with the momentum changes in the flow. Momentum theory is then applied to determine the axial and tangential induced velocities at the rotor plane. It has been argued that only the lift force should be included in the momentum balancing, excluding the drag force (Wilson and Lissaman, 1974; Branlard, 2017). Furthermore, the radial induced velocity is not accounted for in the momentum theory and is not standard in BEM implementations (Madsen et al., 2020).

Unlike conventional BEM literature and textbooks, we formulate the BEM method in this section using vortex theory, which has been shown to be equivalent (Branlard and Gaunaa, 2015a; Li et al., 2025b). The Kutta–Joukowski thrust and power coefficients are derived from the projection of the lift force, assuming zero radial induction (Li et al., 2025b):

$$C_{t,\mathrm{KJ}}^{\mathrm{pl}} = k_s(1+a'),\tag{1}$$

$$C_{p,KJ}^{\text{pl}} = k_s(1 - a_B),$$
 (2)

where  $k_s$  is the normalized circulation strength:

$$k_s = \frac{\Omega\Gamma}{\pi U_0^2}. (3)$$

Neglecting the radial induction results in these coefficients being identical to those of a corresponding planar rotor with straight blades and have the same circulation distribution (Li et al., 2025b), as indicated by the superscript "pl".

The bound circulation of the blade is calculated from the circulatory part of the lift coefficient  $C_L^C$ . Assuming all blades are operating under the same conditions, the total bound circulation strength is:

$$\Gamma = N_B \Gamma_B = \frac{1}{2} N_B V_{\text{rel}} c C_L^C. \tag{4}$$

The tangential induction can be derived either by balancing the angular momentum (Madsen et al., 2020) using Eq. (2) or directly calculated from the circulation  $\Gamma$ , as in the vortex cylinder model (Øye, 1990; Branlard and Gaunaa, 2015b; Li et al., 2022b):

$$a' = \frac{\Gamma}{4\pi\Omega r^2}$$
. (5)

Prandtl's tip-loss correction is commonly applied in BEM-based methods to account for the influence of a finite number of blades on the axial induction (Glauert, 1935; Sørensen, 2015). This correction increases the blade's axial induction compared to the annulus-averaged value when moving towards the tip. Various implementations of the tip-loss correction exist. In this study, we adopt the approach proposed by Madsen et al. (2020):

$$a_B = a_{\text{BEM}} = f_{a-C_t}(C_t^{\text{pl}}/F).$$
 (6)

In the present work, the empirical relationship between the axial induction factor and the thrust coefficient based on the polynomial function by Madsen et al. (2020) is adopted, which is the default relationship in the HAWC2 code (Larsen and Hansen, 2007).

One key advantage of the BEM method is its computational efficiency. However, as demonstrated by Li et al. (2025b, 2024), with a prescribed bound circulation, the BEM method predicts the same inductions and approximately the same loads, which is independent of geometric modifications to the blade (e.g., sweep or prebend), providing these changes do not affect the blade radius. This is because the method does not account for the influence of specific wake geometry on inductions, with the wake corresponding to that of straight blades forming a planar rotor. Nonetheless, the BEM method remains a valuable baseline model and starting point upon which more advanced engineering aerodynamic models can be developed.

## 110 3 Existing coupled near and far wake model for swept blades

The coupled near and far wake model, a hybrid of the lifting line method and the momentum theory, was developed to mitigate the dependency on Prandtl's empirical tip-loss correction used in models based on the actuator disc theory. Additionally, the coupled model provides a more physically consistent framework for modeling unsteady tip-loss effects. Since its first application in wind turbine modeling (Madsen and Rasmussen, 2004), several improvements have been introduced (Pirrung et al., 2014, 2016, 2017a, b), enabling time-domain aeroelastic simulations.

#### 3.1 Background

In this coupled model, the rotor's trailed wake is divided into two components: the *near wake*, which captures immediate effects close to the blade, and the *far wake*, which accounts for wake effects further downstream. The near wake, defined as the first

quarter revolution of the blade's own trailed vortices, is modeled using a simplified lifting line approach with non-expanding helical vortex filaments. An indicial function approach is used for time stepping, with the steady-state induction approximated using empirical equations (Beddoes, 1987) and correction factors (Pirrung et al., 2016, 2017b). This method maintains low computational effort, independent of the elapsed simulation time.

The remaining trailed vortices, including those of the blade itself and the other blades, are defined as the far wake. In the original coupled model, the far wake is modeled based on momentum theory (Madsen and Rasmussen, 2004). A rotor-averaged coupling factor  $k_{\rm FW}^{\rm MT}$  is applied to scale down the far wake inductions relative to the full trailed wake inductions:

$$a_{\text{FW, MT}} = f_{a-C_t}(k_{\text{FW}}^{\text{MT}}C_t), \tag{7}$$

$$a'_{\text{FW.MT}} = k_{\text{FW}}^{\text{MT}} a'. \tag{8}$$

While this direct scaling approach is straightforward, it lacks physical consistency because it uniformly reduces the far wake induction relative to the full trailed wake induction, rather than capturing the spanwise variation due to the actual far wake geometry and the cross-talk between annular sections.

The total axial and tangential inductions are obtained by summing the near wake and far wake contributions. The radial induction is not included, as it is also excluded in standard BEM implementations (Madsen et al., 2020):

$$a_{B,\text{tot}} = a_{\text{NW}} + a_{\text{FW}},\tag{9}$$

$$a'_{\text{tot}} = a'_{\text{NW}} + a'_{\text{FW}}.$$
 (10)

# 135 3.2 Modeling of swept blades

To accurately predict the aerodynamic loads of swept blades, modifications to the coupled model are necessary to account for the influence of blade sweep on inductions. Previous studies (Li et al., 2018, 2022d; Fritz et al., 2022, 2024b) have identified two primary effects of blade sweep on inductions. First, the blade's bound vortex, which is assumed following the 1/4 chord line of the blade, becomes curved and has an influence on itself (Li et al., 2018, 2020). Second, the starting position of the trailed vortex follows the swept blade geometry and is thus shifted within the rotor plane compared to a straight blade. The bound and trailed vortices of a backward swept blade are illustrated in Fig. 1.

The trailing functions of a swept blade, representing the contributions of an elementary trailed vortex to inductions as a function of the azimuthal angle  $\beta$ , are derived. The position vectors x and the elementary trailed vortex ds are used to determine the trailing functions in the blade local coordinate system (BL-sys):

$$x = \begin{pmatrix} R\sin(\beta + \psi) \\ -R\beta\tan\varphi \\ R - h - R\cos(\beta + \psi) \end{pmatrix}$$
, (11)

$$ds = R d\beta \begin{pmatrix} -\cos(\beta + \psi) \\ \tan \varphi \\ -\sin(\beta + \psi) \end{pmatrix}, \tag{12}$$

Figure 1. Front view of a backward swept wind turbine blade rotating clockwise, in the blade local coordinate system (BL-sys). The rotational speed is  $\Omega$  and the free wind speed is  $U_0$ . The radius of the trailing point (tp) is R and the radius of the calculation point (cp) is r, with a difference in radius of h. The sweep angle  $\psi$  is defined as the azimuthal difference between tp and cp. The azimuthal difference between the elementary trailed vorticity ds and the tp is  $\beta$ . The position vector x points from ds to the cp. The figure is adapted from (Li et al., 2022d) with updated coordinate system definitions detailed in (Li et al., 2025b).

where h is defined as the difference in radius between the trailing point and the calculation point:

$$h \equiv R - r,\tag{13}$$

and the helix angle  $\varphi$  of the trailed vortex is calculated from the inductions at the trailing point (tp) using:

$$\varphi \equiv \frac{U_0(1 - a_{B,\text{tot}})}{\Omega R(1 + a'_{\text{tot}})}.$$
 (14)

For a trailed vortex element ds with a strength of  $\Delta\Gamma_B$ , the elementary induced velocity at the blade section is calculated according to the Biot-Savart law:

$$d\mathbf{w} = -\frac{\Delta\Gamma_B}{4\pi} \frac{\mathbf{x} \times d\mathbf{s}}{\|\mathbf{x}\|^3}.$$
 (15)

The trailing functions of axial, tangential and radial induced velocities, corresponding to the y-, x- and z-components of dw, are expressed as:

$$dw_y = -\frac{\Delta\Gamma_B ds \cos\varphi}{4\pi R^2} \frac{1 - \tilde{r}\cos(\beta + \psi)}{\left[1 + \tilde{r}^2 - 2\tilde{r}\cos(\beta + \psi) + (\beta\tan\varphi)^2\right]^{\frac{3}{2}}},\tag{16}$$

$$dw_x = \frac{\Delta\Gamma_B ds \sin\varphi}{4\pi R^2} \frac{\tilde{r} - \cos(\beta + \psi) - \beta \sin(\beta + \psi)}{\left[1 + \tilde{r}^2 - 2\tilde{r}\cos(\beta + \psi) + (\beta \tan\varphi)^2\right]^{\frac{3}{2}}},\tag{17}$$

$$dw_{y} = -\frac{\Delta\Gamma_{B} ds \cos\varphi}{4\pi R^{2}} \frac{1 - \tilde{r}\cos(\beta + \psi)}{\left[1 + \tilde{r}^{2} - 2\tilde{r}\cos(\beta + \psi) + (\beta\tan\varphi)^{2}\right]^{\frac{3}{2}}},$$

$$dw_{x} = \frac{\Delta\Gamma_{B} ds \sin\varphi}{4\pi R^{2}} \frac{\tilde{r} - \cos(\beta + \psi) - \beta\sin(\beta + \psi)}{\left[1 + \tilde{r}^{2} - 2\tilde{r}\cos(\beta + \psi) + (\beta\tan\varphi)^{2}\right]^{\frac{3}{2}}},$$

$$dw_{z} = \frac{\Delta\Gamma_{B} ds \sin\varphi}{4\pi R^{2}} \frac{-\sin(\beta + \psi) + \beta\cos(\beta + \psi)}{\left[1 + \tilde{r}^{2} - 2\tilde{r}\cos(\beta + \psi) + (\beta\tan\varphi)^{2}\right]^{\frac{3}{2}}},$$
(18)

where:

$$ds = V_{\text{rel}} dt = R \frac{d\beta}{\cos \varphi} \equiv R d\beta^*,$$
 (19)

$$\tilde{r} \equiv \frac{r}{R} = 1 - \frac{h}{R}.\tag{20}$$

Integrating these trailing functions with respect to the azimuthal angle  $\beta$  from 0 to  $\pi/2$  gives the steady-state near wake inductions:

$$W = \int_{\beta=0}^{\beta=\frac{\pi}{2}} \mathrm{d}w. \tag{21}$$

In the existing near wake model, the radial induction due to the trailed wake is neglected. The trailing functions of the axial and tangential inductions in Eqs. (16) and (17) are approximated by the sum of two exponential functions. Following Beddoes (1987), the parameters  $A_1 = 1.359$ ,  $A_2 = -0.359$ ,  $b_1 = 1$  and  $b_2 = 4$  are chosen.

$$d\tilde{w} = \frac{\Delta\Gamma_B R}{4\pi h|h|} (A_1 e^{-b_1 \beta^*/\Phi} + A_2 e^{-b_2 \beta^*/\Phi}) d\beta^*$$
(22)

By integrating the approximated trailing function from zero to infinity, the approximated steady-state near wake induction is obtained:

$$\tilde{W} = \frac{\Delta \Gamma_B R}{4\pi h |h|} \left( \frac{A_1}{b_1} + \frac{A_2}{b_2} \right) \Phi \approx W. \tag{23}$$

According to Eq. (23), the steady-state induction can be represented using  $\Phi$ , which is a function of the radial position  $\tilde{r}$ , sweep angle  $\psi$  and helical angle  $\varphi$ . Direct numerical integration of the trailing functions from the Biot-Savart law would be computationally intensive. To address this, previous modifications to the near wake model use steady-state inductions at special conditions and apply corrections for general conditions (Li et al., 2022d). First, the steady-state inductions for the special conditions of  $\varphi=0$  and  $\varphi=\pi/2$ , corresponding to an in-plane trailed vortex and a straight trailed vortex convecting downstream, are derived analytically and labeled as  $\Phi_{ip}$  and  $\Phi_{ss}$ , respectively. Then, the steady-state inductions for general helical angles are calculated using these special conditions and a correction factor  $k_{\Phi}$ , based on pre-calculated influence coefficients.

$$\Phi = f_{conv}(k_{\Phi}, \Phi_{ip}, \Phi_{ss}) \tag{24}$$

This approach, as detailed in (Li et al., 2022d), has relatively high accuracy and maintains low computational effort. It is applied to the near wake axial and tangential inductions, while the radial induction is neglected. The total near wake axial and tangential inductions also include the influence of the curved bound vortex:

$$a_{\text{NW}} = a_{\text{bound}} + a_{\text{NW, trail}},\tag{25}$$

$$a'_{\text{NW}} = a'_{\text{bound}} + a'_{\text{NW, trail}}.$$
(26)

It is assumed that the sweep effect is fully captured by the blade's own near wake trailed vortex and the curved bound vortex influence, while the influence of the other blade's bound vorticity and near wakes is neglected. As a result, the far wake does

not have to account for sweep effects. Consequently, the same far wake model as for straight blades, based on momentum theory and described in Sect. 3.1, can be applied. The total axial and tangential inductions are then the sum of the near wake and far wake contributions. For the radial induction, only the influence of the bound vortex is included.

$$a_{B,\text{tot}} = a_{\text{NW}} + a_{\text{FWMT}} \tag{27}$$

$$a'_{\text{tot}} = a'_{\text{NW}} + a'_{\text{FWMT}} \tag{28}$$

$$u_{r,\text{tot}} = u_{r,\text{bound}}$$
 (29)

This modified coupled model is labeled as NW-MT, representing the coupling of the near wake model and the far wake momentum theory. For different swept blades operating under optimal conditions, loads predicted by NW-MT were compared with results from higher-fidelity aerodynamic models, such as a free wake lifting line method and a RANS CFD solver. Results from NW-MT show significant improvements over the BEM method in predicting sweep effects (Li et al., 2022d).

## 4 Far wake vortex cylinder model

This section introduces the far wake vortex cylinder model as a replacement for the momentum theory-based far wake model in the coupled model. This approach provides a more physically consistent representation of the far wake.

#### 200 4.1 Elementary right vortex cylinder

First, the elementary right vortex cylinder, a fundamental element of the far wake vortex cylinder model, is introduced. The trailed vortex of a uniformly loaded planar rotor can be represented using a non-expanding vortex cylinder (Branlard and Gaunaa, 2015a). Assuming the rotor operates under uniform inflow with zero yaw error and no rotor tilt, the vortex cylinder will be a right cylinder rather than an oblique one (Branlard and Gaunaa, 2015b). The cylindrical vortex sheet is decomposed into tangential and longitudinal components, denoted as  $\gamma_t$  and  $\gamma_t$ , respectively.

For a vortex cylinder with radius R, the axial, tangential and radial induced velocities at a point with radius r and axial position y (positive downstream from the rotor disc) are given by:

$$u_a(r,y) = \frac{\gamma_t}{2} \left[ \frac{R-r+|R-r|}{2|R-r|} + \frac{y\sqrt{m}}{2\pi\sqrt{rR}} \left( K(m) + \frac{R-r}{R+r} \Pi(n,m) \right) \right] \equiv \tilde{u}_a(r,R,y)\gamma_t, \tag{30}$$

$$u_{t,l}(r,y) = \frac{\gamma_l}{2} \frac{R}{r} \left[ \frac{r - R + |R - r|}{2|R - r|} + \frac{y\sqrt{m}}{2\pi\sqrt{rR}} \left( K(m) - \frac{R - r}{R + r} \Pi(n,m) \right) \right] \equiv \tilde{u}_{t,l}(r,R,y)\gamma_l, \tag{31}$$

$$u_r(r,y) = -\frac{\gamma_t}{2\pi} \sqrt{\frac{R}{r}} \left[ \frac{2-m}{\sqrt{m}} K(m) - \frac{2}{\sqrt{m}} E(m) \right] \equiv \tilde{u}_r(r,R,y) \gamma_t,$$
 (32)

where:

$$m = \frac{4rR}{(R+r)^2 + y^2},\tag{33}$$

$$n = \frac{4rR}{(R+r)^2},\tag{34}$$

and K(m), E(m) and  $\Pi(n,m)$  are the complete elliptic integrals of the first, second and third kind, respectively.

# 215 4.2 Superposition of far wake vortex cylinders

To model the radially varying bound circulation of a rotor, concentric vortex cylinders with different radii and circulations are superpositioned (Branlard and Gaunaa, 2015a). This approach models the full trailed wake and is capable of capturing the rotor's non-planar effects (Li et al., 2022b).

In this study, vortex cylinders are used to model the far wake, replacing the existing far wake momentum model in the coupled model. This provides a physically more consistent representation of the far wake, addressing the limitations discussed in Sect. 3.1. The far wake vortex cylinders are positioned a constant distance  $\Delta y_{\rm FW}$  downstream from the rotor plane<sup>1</sup>, as illustrated in Fig. 2, for both a planar rotor (panel a) and a non-planar rotor (panel b). The concept of using a constant downstream offset distance aligns with applying a rotor-averaged coupling factor for the entire rotor in the far wake momentum theory. The far wake vortex cylinders can be considered as the far wake part of the full wake vortex cylinders, which effectively subtracts the near wake part from the full trailed wake. This decomposition is also illustrated in Fig. 2.

Figure 2. Decomposition of the full trailed wake vortex cylinder into a near wake part and a far wake part for: (a) a planar rotor; (b) a non-planar rotor. All far wake vortex cylinders begin a constant distance of  $\Delta y_{\rm FW}$  further downstream from the rotor plane, which is the surface swept by the blades.

## 4.2.1 Definition of coupling factor

For the far wake vortex cylinder model, a rotor-averaged coupling factor  $k_{\rm FW}^{\rm VC}$  is defined. The near wake is assumed to convect downstream at a constant velocity that is evaluated at the blade. The corresponding near wake helical pitch  $h_{\rm NW}$  is:

$$h_{\text{NW}} = \frac{2\pi U_0 (1 - a_{B, \text{tot}})}{\Omega (1 + a'_{\text{tot}})} = 2\pi r \tan \varphi.$$
 (35)

First, the near wake helical pitch is weighted-averaged by the annulus area to obtain the rotor-averaged value  $\bar{h}_{\rm NW}$ :

$$\bar{h}_{\text{NW}} = \frac{\sum_{i=1}^{N_{\text{cp}}} h_{\text{NW},i} A_{a,i}}{\sum_{i=1}^{N_{\text{cp}}} A_{a,i}},\tag{36}$$

<sup>&</sup>lt;sup>1</sup>In this work, the rotor plane refers to the surface swept by the blades.

where  $A_{a,i}$  is the annulus area at the *i*-th calculation point (cp), calculated from the radii of the surrounding trailing points (tp):

$$A_{a,i} = \pi(R_{i+1}^2 - R_i^2). \tag{37}$$

Since the near wake is defined as the first quarter revolution of the trailed vortex, the downstream distance traveled by the near wake corresponds to one-fourth of  $\bar{h}_{\rm NW}$ . This distance is further scaled by the coupling factor  $k_{\rm FW}^{\rm VC}$ , which is limited to values between 0 and 1, reflecting the decreasing convection speed of the trailed wake as it convects downstream<sup>2</sup>.

$$\Delta y_{\rm FW} = \frac{1}{4} k_{\rm FW}^{\rm VC} \bar{h}_{\rm NW} \tag{38}$$

The axial coordinate of a far wake vortex cylinder at radius  $R_j$  is calculated by shifting the trailing point's axial coordinate downstream by  $\Delta y_{\rm FW}$ :

$$y_{\text{FW},j} = y(R_j) + \Delta y_{\text{FW}}. \tag{39}$$

#### 4.2.2 Calculation of far wake induction

Each far wake vortex cylinder is assumed to convect downstream at a constant speed, which is the mean value of the two velocities just outside and inside the vortex sheet, at infinitely far downstream. Since the same assumption is also used for the system closure of full vortex cylinders (Branlard and Gaunaa, 2015a), the far wake vortex cylinder system can be solved using the same method (Li et al., 2022b). First, the annulus-averaged axial inductions of the corresponding planar full vortex cylinders are calculated similarly to Eq. (6) but neglecting the tip-loss correction:

$$a_{\infty}^{\rm pl} = f_{a-C_{\star}}(C_{\star}^{\rm pl}),$$
 (40)

where the subscript  $\infty$  denotes the annulus-averaged value.

Then, the tangential vorticity is obtained from the difference in annulus axial induction factors of the two neighbouring sections:

$$\gamma_{t,j} = 2U_0(a_{\infty,j+1}^{\text{pl}} - a_{\infty,j}^{\text{pl}}). \tag{41}$$

The strength of the longitudinal vortex sheet for the vortex cylinder with radius  $R_j$  and total trailed circulation  $\Delta\Gamma_j$  is:

$$\gamma_{l,j} = \frac{\Delta \Gamma_j}{2\pi R_j}.\tag{42}$$

<sup>&</sup>lt;sup>2</sup>The coupling factor needs to be smaller than one also because the far wake includes the trailed wake contributions from the two other blades

For a calculation point at radius r and axial coordinate y, the contributions from all far wake vortex cylinders are summed to obtain the far wake inductions:

$$a_{\text{FW,VC}} = -\frac{1}{U_0} \sum_{i=1}^{N_{\text{tp}}} \tilde{u}_a \left( r, R_j, y - y_{\text{FW}, j} \right) \gamma_{t, j}, \tag{43}$$

$$a'_{\text{FW,VC}} = -\frac{1}{\Omega r} \sum_{j=1}^{N_{\text{tp}}} \tilde{u}_{t,l} \left( r, R_j, y - y_{\text{FW},j} \right) \gamma_{l,j}, \tag{44}$$

$$u_{r,\text{FW,VC}} = \sum_{j=1}^{N_{\text{tp}}} \tilde{u}_r \left( r, R_j, y - y_{\text{FW},j} \right) \gamma_{t,j}. \tag{45}$$

In the present work, only blade sweep effects are included and non-planar effects are neglected, assuming both the full wake vortex cylinders and the far wake vortex cylinders are planar, as depicted in panel (a) of Fig. 2. Therefore, the axial coordinate difference, which is the last term in the brackets of Eqs. (43) to (45), simplifies to:

$$y - y_{\text{FW},j} = -\Delta y_{\text{FW}}.\tag{46}$$

## 5 Modified coupled near and far wake models

Building upon the limitations identified in the existing coupled near and far wake model (NW-MT), this section introduces three modified coupled models aimed at improving aerodynamic load predictions for swept blades. The first two methods use the far wake vortex cylinder model, while the third corrects the existing NW-MT model by including the trailed wake radial induction from the vortex cylinder model.

## 5.1 Idealized near wake model coupled with far wake vortex cylinder

The first modified coupled model, labeled as NW(ideal)-VC, is introduced to provide a benchmark of the highest fidelity that can be achieved within the coupled framework. It extends the existing NW-MT model by using an idealized near wake model that captures sweep effects with the highest possible fidelity, without regard to computational cost or model complexity. Additionally, it replaces the far wake momentum theory with the more physically consistent far wake vortex cylinder model introduced in Sect. 4.2.

The near wake is defined as the first quarter revolution of the blade's own trailed wake, following the original coupled model.

The near wake trailed vortex inductions are calculated from the numerical integration of the trailing functions in Eqs. (16) to (18) derived from the Biot–Savart law, according to Eq. (21). The near wake inductions are defined as the sum of the inductions from the bound vortex and the near wake trailed vortex of the own blade:

$$a_{\text{NW}} = a_{\text{bound}} + a_{\text{NW. trail}}^{\text{num}},\tag{47}$$

$$a'_{NW} = a'_{bound} + a'_{NW, trail}^{num},$$
 (48)

$$u_{r,\text{NW}} = u_{r,\text{bound}} + u_{r,\text{NW, trail}}^{\text{num}},\tag{49}$$

where the superscript "num" indicates that the inductions are calculated from numerical integration.

The far wake inductions are calculated from the far wake vortex cylinder model using Eqs. (43) to (45). The total inductions are then the sum of the near wake and far wake contributions:

$$a_{B,\text{tot}} = a_{\text{NW}} + a_{\text{FW,VC}},$$
 (50)

$$a'_{\text{tot}} = a'_{\text{NW}} + a'_{\text{FWVC}},\tag{51}$$

$$u_{r,\text{tot}} = u_{r,\text{NW}} + u_{r,\text{FW,VC}}.\tag{52}$$

This idealized NW(ideal)-VC method includes the influence of a finite number of blades on axial, tangential and also radial inductions. However, this model is computationally expensive, as the near wake inductions require numerical integration at each iteration.

# 5.2 Existing near wake model coupled with far wake vortex cylinder

The second modified coupled model, labeled as NW-VC, is introduced as the primary model targeted in this work, offering a balance between fidelity and computational efficiency. It addresses the computational challenges of NW(ideal)-VC by simplifying the near wake model while still improving upon NW-MT through the use of the far wake vortex cylinder model.

The near wake axial and tangential inductions are calculated using the computationally efficient approach as in NW-MT, using Eqs. (25) and (26). The far wake axial and tangential inductions are obtained from the far wake vortex cylinder model, using Eqs. (43) and (44). The total axial and tangential inductions are then the sum of the near wake and far wake contributions, following the same forms as Eqs. (50) and (51) for NW(ideal)-VC. For blades with only sweep and no prebend, the error in near wake axial and tangential inductions in NW-VC is shown to be small compared to direct numerical integration as in NW(ideal)-VC (Li et al., 2022d).

The trailed wake radial induction, including both near wake and far wake contributions, is modeled using the full trailed wake vortex cylinder model:

$$u_{r,\text{VC}} = \sum_{j=1}^{N_{\text{tp}}} \tilde{u}_r \left( r, R_j, y - y(R_j) \right) \gamma_{t,j}.$$
 (53)

Under the assumption of a planar rotor without tilt or coning, the axial coordinate differences simplify:

$$y - y(R_j) = 0.$$
 (54)

The total radial induction includes contributions from both the bound vortex and the trailed vortex:

$$u_{r,\text{tot}} = u_{r,\text{bound}} + u_{r,\text{VC}}.$$
 (55)

Unlike the idealized NW(ideal)-VC model that models the near wake trailed vortex radial induction using non-expanding helical vortex filaments and adds it to the far wake vortex cylinder induction, the simplified NW-VC model uses the vortex cylinder model for the radial induction from the entire trailed wake. As a result, the influence of a finite number of blades on the radial induction is not included in NW-VC.

# 5.3 Radial induction correction to NW-MT using vortex cylinder model

The third modified coupled model, labeled as NW-MT-VC, serves as a comparison to the previous NW-MT model to assess the impact of radial induction on aerodynamic loads of swept blades. It modifies the existing NW-MT model by including the trailed wake radial induction calculated from the full trailed wake using the vortex cylinder model.

The axial and tangential inductions are calculated identically to NW-MT, using Eqs. (27) and (28). The radial induction is calculated the same as in NW-VC, using Eq. (55).

# 5.4 Comparison between coupled models

Table 1 summarizes the relationships between the four different coupled models, listed in ascending order of fidelity.

**Table 1.** Comparison between different coupled models, in ascending order of fidelity. The modeling of axial, tangential and radial induction components are compared. NW-MT represents the previous implementation; NW-MT-VC extends this by including radial induction from the vortex cylinder (VC) model; NW-VC further improves the far wake modeling and is the primary model; Finally, NW(ideal)-VC serves as an idealized reference model.

| Model        | Near wake $(a_{NW}, a'_{NW})$   | Far wake $(a_{FW}, a'_{FW})$ | Radial induction $(u_r)$                    |
|--------------|---------------------------------|------------------------------|---------------------------------------------|
| NW-MT        | Approximate, Eqs. (25) and (26) | FW MT, Eqs. (7) and (8)      | Bound vortex, Eq. (29)                      |
| NW-MT-VC     | Approximate, Eqs. (25) and (26) | FW MT, Eqs. (7) and (8)      | Bound vortex + VC, Eq. (55)                 |
| NW-VC        | Approximate, Eqs. (25) and (26) | FW VC, Eqs. (43) and (44)    | Bound vortex + VC, Eq. (55)                 |
| NW(ideal)-VC | Exact, Eqs. (47) and (48)       | FW VC, Eqs. (43) and (44)    | Bound vortex + NW trailed + FW VC, Eq. (52) |

From NW-MT to NW-MT-VC, the radial induction due to the trailed wake is included, modeled using the vortex cylinder model. From NW-MT-VC to NW-VC, the far wake axial and tangential inductions are calculated using the more physically consistent far wake vortex cylinders, instead of the far wake momentum theory. Finally, from NW-VC to NW(ideal)-VC, the radial induction due to the near wake trailed vortex is modeled using non-expanding helical vortex filaments, instead of vortex cylinders. Hence, the influence of a finite number of blades on the radial induction is included. In addition, the near wake axial and tangential inductions are calculated using numerical integration, resulting in slightly improved accuracy when modeling blades with only sweep and no prebend.

In summary, the proposed modified coupled models address the limitations of previous aerodynamic models for swept blades. These models provide a framework for evaluating the impact of various modeling choices on aerodynamic load predictions, which will be analyzed in the subsequent sections.

#### 6 Investigation and improvement of the coupling factor

Previous work (Li et al., 2022d) demonstrated that the existing automatically-adjusted coupling factor in the coupled NW-MT model performs well for straight and backward swept blades but exhibits unsatisfactory performance for forward swept

blades. Although the primary source of the issue was narrowed down to the coupling factor, the underlying causes were not analyzed in depth. This section provides a detailed examination of the current coupling factor to identify the reasons for its insufficient performance. Modifications are then proposed to address these issues, aiming to enable a more robust, automatically-adjusted coupling factor for general swept blades. Improving the coupling factor will enhance the accuracy and reliability of aerodynamic load predictions for swept blades, which is critical for blade design optimizations. Because coupling factors are calculated very similarly for both the far wake momentum theory ( $k_{\rm FW}^{\rm MT}$ ) and the far wake vortex cylinder model ( $k_{\rm FW}^{\rm VC}$ ), NW-MT is used for this investigation<sup>3</sup>. The blades used for the investigation are the baseline straight blade (Str), the backward swept blade (mB-1) and the forward swept blade (mB-5), as described in Sect. 7.3.

# 6.1 Existing coupling factor

The existing coupling factor, which is adjusted automatically at each time step as described by Pirrung et al. (2016), was initially implemented in HAWC2 version 12.3, released in 2016. In HAWC2 version 12.4 (released in 2017), the rotor-averaged coupling factor was modified to be weighted by annulus area instead of thrust force, to improve numerical stability in aeroelastic simulations. This modified approach is described below.

First, local coupling factors are introduced for each blade section. For section i, the local coupling factor at time step t, labeled as  $\kappa_{\text{FW},i}^t$ , is updated from the previous time step using the Newton–Raphson method<sup>4</sup>:

$$\kappa_{\text{FW},i}^t = \kappa_{\text{FW},i}^{t-1} - \frac{\delta a_i}{\partial \delta a_i / \partial k_{\text{FW}}},\tag{56}$$

where  $\delta a$  represents the difference between the total axial induction from the coupled model and a reference BEM model that has tip-loss correction applied:

$$\delta a = a_{B,\text{tot}} - a_{\text{BEM}} = a_{\text{NW}} + a_{\text{FW}} - a_{\text{BEM}}.$$
(57)

Since the near wake induction  $a_{\rm NW}$  and the reference BEM induction  $a_{\rm BEM}$  are not direct functions of the coupling factor, the partial derivative  $\partial \delta a/\partial k_{\rm FW}$  is derived as:

$$\frac{\partial \delta a}{\partial k_{\rm FW}} = \frac{\partial a_{\rm FW}}{\partial k_{\rm FW}}.$$
 (58)

For the far wake momentum theory, the far wake inductions are radially independent, so this partial derivative can be directly derived from Eq. (7), resulting in simple expressions (Pirrung et al., 2016). In contrast, in the far wake vortex cylinder model, contributions from all vortex cylinders affect the far wake induction of each section. Hence, the derivations are more complex, as detailed in Appendix B.

To limit the impact of extreme values caused by strong trailed vortices, the sectional coupling factor is limited to values between 0 and 1, denoted as  $\tilde{\kappa}_{FW}$ . The rotor-averaged coupling factor is then obtained by weighting these limited local coupling

 $<sup>\</sup>overline{\phantom{a}}^3$ Insights gained from analyzing  $k_{\rm FW}^{\rm MT}$  used in NW-MT are expected to be directly transferable to  $k_{\rm FW}^{\rm VC}$  used in the far wake vortex cylinder model, making NW-MT a suitable and sufficient choice for identifying the issues.

<sup>&</sup>lt;sup>4</sup>In practice, the rotor-averaged value  $k_{\rm FW}^{t-1}$  at the previous time step (t-1) is used for the calculation, instead of  $\kappa_{{\rm FW},i}^{t-1}$ .

factors by the annulus area  $A_{a,i}$ :

$$k_{\text{FW}} = \frac{\sum_{i=1}^{N_{\text{cp}}} \tilde{\kappa}_{\text{FW},i} A_{a,i}}{\sum_{i=1}^{N_{\text{cp}}} A_{a,i}}.$$
(59)

In the far wake momentum theory, this rotor-averaged coupling factor  $k_{\rm FW}$  is applied uniformly across all rotor sections to calculate the far wake induction, as shown in Eqs. (7) and (8). Similarly, in the far wake vortex cylinder model, all vortex cylinders are shifted downstream by a constant distance that is scaled by  $k_{\rm FW}$ , as shown in Eq. (38). Importantly, the sectional coupling factor  $\kappa_{\rm FW}$  is not used directly in the far wake model, since it would cause the converged axial induction to tend towards the value from the reference BEM method for each section (Pirrung et al., 2016)<sup>5</sup>.

## **6.1.1** Performance of the existing coupling factor

For blades without significant sweep, the coupled NW-MT model with the existing coupling factor has been extensively used for aeroelastic simulations and has generally shown satisfactory performance and numerical stability across various operational conditions(Pirrung et al., 2017a, b). However, for blades with significant forward sweep, the existing coupling factor for the NW-MT model shows insufficient performance.

Numerical tests were performed to evaluate the steady-state loads for both backward and forward swept blades with different geometries at the optimal operational condition (Li et al., 2022d). It is expected that both the rotor-averaged coupling factor and far wake inductions for the swept blades should closely match those of the straight blade, as the blade sweep effect is assumed to be fully captured in the near wake part of the coupled model. The current method of calculating the automatically-adjusted coupling factor performs well for the backward swept blades tested in the previous work (Li et al., 2022d), predicting similar coupling factor values as the baseline straight blade. However, for all forward swept blades tested in that previous work, the automatically-adjusted coupling factor consistently exhibits noticeable underestimations, leading to overestimated loads across the blade span. Using a fixed coupling factor equal to that of the baseline straight blade was found to show good agreement with higher-fidelity models for all tested swept blades. This suggests that the modeling of bound and near wake trailed vortices of swept blades is sufficient. However, a fixed coupling factor is not feasible for general aeroelastic calculations, as the rotor load and hence the coupling factor vary over time.

# 6.2 Analysis of the existing coupling factor

To identify the issues with the existing method of calculating the coupling factor, an investigation was performed using the NW-MT model. At the optimal operational condition with a wind speed of 8 m s<sup>-1</sup>, different components of the axial inductions and the local coupling factor  $\tilde{\kappa}_{FW}$  along the blade span are compared, as depicted in Fig. 3. Three different values of axial induction are compared: the near wake induction  $a_{NW}$  in Eq. (25), the far wake induction  $a_{FW,MT}$  from the far wake momentum theory in Eq. (7) and the difference between the total axial induction and the reference BEM induction, denoted as  $\delta a$  in Eq. (57).

<sup>&</sup>lt;sup>5</sup>Except the sections where the sectional coupling factor κ<sub>FW</sub> would fall outside of the range from 0 to 1 without limiting.

Figure 3. Comparison of: (a) local coupling factor  $\tilde{\kappa}_{FW}$ ; (b) different components of axial induction factors from the NW-MT(origin) model with the existing coupling factor, for the baseline straight blade Str, backward swept blade mB-1 and forward swept blade mB-5, at a wind speed of 8 m s<sup>-1</sup>.

It is observed that from the blade root to the mid-span (radius of 50 m), the local coupling factor exhibits only small differences between different blades, which have minimal impacts on the rotor-averaged value. Therefore, the focus is on the outer half of the blade, from the mid-span to the blade tip.

For the baseline straight blade (Str), as shown in Fig. 3 (a), the local coupling factor  $\tilde{\kappa}_{FW}$  slowly decreases from mid-span to a radius of 90 m, maintaining a high level of approximately 0.9. Beyond 90 m,  $\tilde{\kappa}_{FW}$  increases steeply and quickly reaches the upper limit of 1.0 at a radius of 95 m, maintaining this upper limit until the blade tip. This behavior of  $\tilde{\kappa}_{FW}$  is directly related to the axial induction difference  $\delta a$ , according to Eq. (56).

As shown in Fig. 3 (b), from a radius of 50 m to 90 m,  $\delta a$  remains around zero and slowly increases. For radii larger than 90 m,  $\delta a$  decreases steeply with increasing radius. This indicates that the axial inductions from NW-MT and BEM are generally similar, except for the blade tip region. Near the blade tip, different methods of modeling tip effects in NW-MT and BEM result in relatively large differences in axial induction. Due to the upper limit of 1.0 applied to the local coupling factor  $\tilde{\kappa}_{\rm FW}$ , the impact of these large differences on the rotor-averaged value is limited. The resulting rotor-averaged coupling factor  $k_{\rm FW}$  is 0.909, closely resembling the overall level of the local coupling factor  $\tilde{\kappa}_{\rm FW}$ .

## 6.2.1 Backward swept blade

For the backward swept blade mB-1, as shown in Fig. 3 (a), the local coupling factor  $\tilde{\kappa}_{FW}$  behaves similarly to the baseline straight blade (Str) from a radius of 50 m to 90 m, changing slowly with the radius and maintaining a high level of around 0.9. Beyond 90 m,  $\tilde{\kappa}_{FW}$  increases steeply and reaches the upper limit of 1.0 at a radius of 93 m, maintaining this upper limit until the tip. For mB-1, the value of  $\tilde{\kappa}_{FW}$  is slightly lower than Str, from a radius of 50 m to 80 m but slightly higher from 80 m to

95 m. For radii larger than 95 m, both Str and mB-1 reach the upper limit. The resulting rotor-averaged coupling factor  $k_{\text{FW}}$  for mB-1 is 0.904, which is very similar to that of Str.

As shown in Fig. 3 (b), the far wake inductions of mB-1 and Str are nearly identical, due to the similar value of rotor-averaged  $k_{\rm FW}$ . Consequently, the difference in  $\delta a$  is dominated by the near wake axial induction  $a_{\rm NW}$ . The near wake model captures the backward blade sweep effects on the bound and near wake trailed vortices, leading to different near wake axial inductions  $a_{\rm NW}$  between mB-1 and Str. For mB-1,  $a_{\rm NW}$  is higher than Str from a radius of 50 m to 80 m and is lower from 80 m to 90 m. Beyond 90 m,  $a_{\rm NW}$  decreases steeply, which is mainly due to the shifted starting position of the trailed wake within the rotor plane. This results in a significant decrease of  $\delta a$  all the way to the blade tip. According to Eq. (56), this steep decrease in  $\delta a$  should correspond to a steep increase in the local coupling factor  $\tilde{\kappa}_{\rm FW}$ . However, at this operational condition,  $\tilde{\kappa}_{\rm FW}$  is already at high levels around 0.9. Further increases of  $\tilde{\kappa}_{\rm FW}$  will quickly reach the upper limit of 1.0, limiting the impact on the rotor-averaged  $k_{\rm FW}$ .

This suggests that the good performance of the existing coupling factor for the backward swept blades tested in (Li et al., 2022d) may relate to the high loading operational condition and the limiting applied to the local coupling factor  $\tilde{\kappa}_{FW}$ . However, under lower loading conditions with lower overall levels of  $\tilde{\kappa}_{FW}$ , performance might worsen. This is because the steep increase in  $\tilde{\kappa}_{FW}$  near the blade tip may not quickly reach the upper limit, resulting in a noticeable increase in rotor-averaged  $k_{FW}$  and, consequently, an underestimation of the loads. As a result, the performance of the existing coupling factor should also be tested for operational conditions with lower loading, as discussed later in Sect. 8.3.1.

## 425 6.2.2 Forward swept blade

For the forward swept blade mB-5, as shown in Fig. 3 (a), the local coupling factor  $\tilde{\kappa}_{FW}$  is slightly higher than the baseline straight blade (Str) from a radius of 50 m to 80 m. In this region, the value of  $\tilde{\kappa}_{FW}$  is changing slowly with the radius and maintaining a high level of approximately 0.9, showing similar behavior to Str. For radii from 80 m to the blade tip,  $\tilde{\kappa}_{FW}$  decreases steeply all the way to the lower limit of zero, significantly lower compared to Str. Moreover, the higher weighting applied at the tip region when calculating the rotor-averaged value further amplifies this effect. The resulting rotor-averaged coupling factor  $k_{FW}$  is 0.838, notably lower than the straight blade's value of 0.909.

The local coupling factor  $\tilde{\kappa}_{FW}$  is influenced by the axial induction difference  $\delta a$ , which has two components. Firstly, the difference in  $a_{NW}$  between mB-5 and Str, which is due to the influence of the blade's forward sweep, is investigated. As shown in Fig. 3 (b), for mB-5,  $a_{NW}$  is slightly lower than Str from a radius of 50 m to 80 m and is slightly higher from 80 m to 90 m. Beyond 90 m,  $a_{NW}$  is significantly higher than Str, which is mainly due to the shifted starting position of the trailed vortex within the rotor plane. Secondly, due to the lowered value of the rotor-averaged  $k_{FW}$ , the far wake induction for mB-5 is lower than Str, showing an offset. Consequently, the overall level of  $\delta a$  for mB-5 is also lower compared to Str, showing a significant offset, except in the very tip region.

This analysis suggests that the insufficient performance of the existing coupling factor for forward swept blades is directly related to the steep decrease in the local coupling factor  $\tilde{\kappa}_{FW}$  near the blade tip. The lower limit of zero applied to  $\tilde{\kappa}_{FW}$  is

insufficient to limit the significant local change. The decrease in the rotor-averaged  $k_{\rm FW}$  results in the total axial induction from NW-MT being underestimated, which in turn leads to an overestimation of the loads.

## 6.2.3 Issues in the existing coupling factor

From the analysis, three major issues with the existing coupling method are identified. Firstly, the Newton–Raphson method is applied to the local coupling factor  $\tilde{\kappa}_{FW}$ , which is not directly applied in the far wake model. Instead, the rotor-averaged value  $k_{FW}$ , which is the area-weighted value calculated from the local coupling factors, is directly applied in the far wake model. This indirect application of the Newton–Raphson method introduces inconsistencies between the local and rotor-averaged coupling factors. Secondly, the significant differences in axial inductions near the blade tip, resulting from different tip modeling methods in NW-MT and BEM, are balanced by adjusting the overall level of the far wake axial induction. However, this method has limited effectiveness in counteracting the significant differences in axial inductions locally at the blade tip. For example, due to the sweep effects being included in the coupled NW-MT model but not in the reference BEM method. This can lead to large changes in the rotor-averaged  $k_{FW}$  and subsequently affecting load predictions. Thirdly, the impact of the large error locally at the blade tip is amplified by the larger weighting applied to the blade tip region when averaging the coupling factor based on the annulus area.

### 455 **6.3** Modifications to the coupling factor

To address the issues identified in the existing method of calculating the coupling factor, two modifications are proposed. These modifications aim to improve the consistency between local and rotor-averaged coupling factors and to reduce the excessive influence of the blade tip region on the coupling factor calculation. Note that both modifications are applicable to  $k_{\rm FW}^{\rm MT}$  and  $k_{\rm FW}^{\rm VC}$ .

# 6.3.1 First modification

The first modification addresses the inconsistency arising from not directly applying the Newton-Raphson method to the rotor-averaged coupling factor used in the far wake induction. To resolve this, the Newton-Raphson method is applied directly to the rotor-averaged coupling factor  $k_{\rm FW}$ , rather than to the local coupling factor  $\tilde{\kappa}_{\rm FW}$ . The residual function  $F_a$  is therefore defined as the sum of the axial induction difference  $\delta a$  weighted by the corresponding annulus area  $A_a$ :

$$F_a = \sum_{i=1}^{N_{\rm cp}} \delta a_i A_{a,i}. \tag{60}$$

The rotor-averaged coupling factor at the current time step t, labeled as  $k_{\rm FW}^t$ , is updated from the value at the previous time step using the Newton-Raphson method:

$$k_{\text{FW}}^t = k_{\text{FW}}^{t-1} - \frac{F_a^t}{\frac{\partial F_a}{\partial k_{\text{FW}}}}.$$
(61)

By substituting Eq. (60), the partial derivative of  $F_a$  with respect to  $k_{\rm FW}$  is given by:

$$\frac{\partial F_a}{\partial k_{\text{FW}}} = \sum_{i=1}^{N_{\text{cp}}} \frac{\partial a_{\text{FW},i}}{\partial k_{\text{FW}}} A_{a,i}. \tag{62}$$

To prevent excessively large differences in axial induction, particularly near the blade tip, a limiting method is introduced. This involves calculating a tentative local coupling factor  $\kappa_{\text{FW}}$  for each section, following a similar approach to Eq. (56):

$$\kappa_{\text{FW},i}^t = k_{\text{FW}}^{t-1} - \frac{\delta a_i}{\partial a_{\text{FW},i}/\partial k_{\text{FW}}}.$$
(63)

This tentative local coupling factor is then limited to the range between 0 and 1, which is labeled as  $\tilde{\kappa}_{FW}$  after the limiting. Then, the adjusted axial induction difference, labeled as  $\delta \tilde{a}$ , is calculated back from the limited local coupling factor  $\tilde{\kappa}_{FW,i}$ :

$$\delta \tilde{a}_i = (k_{\text{FW}} - \tilde{\kappa}_{\text{FW},i}) \frac{\partial a_{\text{FW},i}}{\partial k_{\text{FW}}}$$
 (64)

This limited axial induction difference  $\delta \tilde{a}$  replaces the original  $\delta a$  in Eq. (60) to calculate the residual function  $F_a$  in Eq. (60). This first modified method of calculating the coupling factor is labeled as method (a).

#### 6.3.2 Second modification

The second modification specifically addresses the impact of significant axial induction differences at the blade tip region, targeting the second and third issues as previously identified. This modification is optional and corresponds to an alternative modified method for calculating the coupling factor.

Inspired by the observation that the bound circulation decreases rapidly towards zero at the blade tip under near-optimal operating conditions, the weighting factor for the rotor-averaged coupling factor is modified. Instead of weighting solely by the annulus area  $A_a$ , the new approach uses the product of  $A_a$  and the normalized bound circulation  $k_s$  as defined in Eq. (3). The residual function is then given by:

$$F_{ka} = \sum_{i=1}^{N_{\rm cp}} k_{s,i} \delta a_i A_{a,i}. \tag{65}$$

The rotor-averaged coupling factor is updated using the Newton–Raphson method:

$$k_{\text{FW}}^t = k_{\text{FW}}^{t-1} - \frac{F_{ka}^t}{\frac{\partial F_{ka}^t}{\partial k_{\text{FW}}^t}}.$$
(66)

By substituting Eq. (65), the partial derivative is obtained using the chain rule:

$$490 \quad \frac{\partial F_{ka}}{\partial k_{\text{FW}}} = \sum_{i=1}^{N_{\text{cp}}} \left( k_{s,i} \frac{\partial a_{\text{FW},i}}{\partial k_{\text{FW}}} + \delta a_i \frac{\partial k_{s,i}}{\partial k_{\text{FW}}} \right) A_{a,i}. \tag{67}$$

Similar to the first modification, the limiting method is applied to prevent excessive influence due to locally large axial induction differences. The tentative local coupling factor  $\kappa_{FW}$  in Eq. (63) is introduced, which is then limited between 0 and

- 1. The adjusted axial induction difference  $\delta \tilde{a}$  that is calculated back using Eq. (64) is then used in Eq. (65), replacing  $\delta a$ , to compute the residual function  $F_{ka}$ .
- This second modified method, labeled as method (ka), effectively reduces the influence of the blade tip region on the coupling factor calculation. So that the rotor-averaged  $k_{\rm FW}$  is expected to better represent the overall level along the blade. Moreover, it can be shown that performing an area-weighted rotor averaging of  $k_s a_B$  is equivalent to averaging of  $C_{p,\rm KJ}^{\rm pl}$  in Eq. (2), which serves as an approximation to the local power coefficient. Additionally, this method is conceptually similar to matching the rotor-averaged axial induction weighted by the annulus thrust force.
- Compared to the first modified method, this second modified method has more complicated forms of the partial derivatives, as shown in Eq. (67). Specifically, computing the partial derivative  $\partial k_s/\partial k_{\rm FW}$  requires additional considerations. The normalized bound circulation  $k_s$  depends on the lift coefficient  $C_L$  and relative velocity  $V_{\rm rel}$  according to Eqs. (3) and (4). The lift coefficient  $C_L$  depends on the angle of attack  $\alpha$  and in turn depends on the induced velocities, which are functions of  $k_{\rm FW}$ . Therefore, the lift slope  ${\rm d}C_L/{\rm d}\alpha$  is required to calculate the partial derivative. This adds complexity to the implementation of this modified method.

## 7 Simulation setup and blades for the comparison

In this section, the higher-fidelity RANS solver and lifting-line (LL) solver are firstly described. Then, the straight and swept blades used for the comparison are described. Finally, loads used for the comparison are described.

# 7.1 Reynolds-averaged Navier-Stokes solver

- The pressure-based incompressible three-dimensional solver EllipSys3D was used to solve the Reynolds-Averaged Navier–Stokes (RANS) equations using finite volume discretization. An inlet/outlet boundary condition strategy was applied at the outer limit of the computational fluid dynamics (CFD) domain. The flow was assumed to be fully turbulent, with the *k*–ω SST model employed (Menter, 1994). Rotor-resolved meshes were generated fully scripted in two consecutive steps to ensure consistent grid quality. First, a structured mesh of the blade surface was generated (Zahle, 2025), with 128 cells in the spanwise direction and 256 cells in the chordwise direction. Second, the surface mesh was radially extruded using a hyperbolic mesh generator (Sørensen, 1998) to create a volume grid with a total of 256 cells. The resulting outer domain was located approximately 11 *D* (rotor diameters) away. A boundary layer clustering was applied with an imposed first cell height of 1 × 10<sup>-6</sup> m to target *y*<sup>+</sup> values lower than one. The resulting volume meshes accounted for 14.2 million cells.
- While a steady solver was employed, some unsteady flow separation is expected near the blade root region during operation.

  Unlike previous works (Li et al., 2022b, c, d), which averaged CFD results over the last 350 iterations, the present work averages only the last 50 iterations, resulting in increased noise in the root region. Applying convergence enhancement methods to the RANS CFD solver, such as the modified BoostConv method (Dicholkar et al., 2022, 2024, 2025), could improve the root region comparisons. However, since the blade root region is not the focus of this study and has negligible influence on the blade tip region, the existing results are deemed sufficient for this study.

# 7.2 Free-wake lifting line solver

In addition to the RANS CFD solver, the free-wake lifting line (LL) module implemented in the in-house multi-fidelity vortex code MIRAS (Ramos-García et al., 2016; Ramos-García et al., 2017) is also used for comparison, as a higher-fidelity aerodynamic model. While the blade-resolved RANS CFD approach resolves the 3-D flow field around the blade geometry, the LL method relies on 2-D airfoil data and applies the cross-flow principle (Hoerner and Borst, 1985). The free-wake LL method represents the highest fidelity within the framework of engineering aerodynamic models that rely on 2-D airfoil data, making it a crucial benchmark for this study.

In the LL method, each blade is represented by a concentrated bound vortex line located at the 1/4 chord line of the blade and is modeled as discrete vortex filaments. With the spanwise and temporal variation of the bound vortex, vorticities are trailed and shed into the wake. Importantly, the influence of a curved bound vortex on itself is included (Li et al., 2020), which is essential for modeling blade sweep effects (Li et al., 2018, 2022d; Fritz et al., 2022, 2024b). The first row of wake vortex panels is released from the lifting line, using velocities locally at the 1/4 chord line. To simulate the wake, a hybrid filament-particle-mesh method is employed. To accurately resolve the near wake, which is crucial for capturing sweep effects, the bound vortex and the first 360 wake panels after each blade are modeled using vortex filaments. This vortex filament wake corresponds to approximately 1.5 revolutions of the rotor<sup>6</sup>. The remaining wake is represented by vortex particles, which are subsequently interpolated onto an auxiliary Cartesian mesh. Interactions between vortex particles and influence of vortex particles on the filaments are efficiently computed using an FFT-based method to solve the Poisson equation with a regularized Green's function under free-space boundary conditions. A 10th-order Gaussian filter is used to regularize the singular free-space Green's function (Hejlesen et al., 2013, 2015). Furthermore, a filter function with a width of 1.5 times of the mesh cell size is applied to minimize smoothing errors. The influence of the vortex filaments on themselves are modeled using the Biot-Savart law. The influence of the vortex filaments on the vortex particles are neglected, since the filament wake size is large enough.

#### 7.2.1 Numerical setup

The Cartesian mesh used for wake modeling spans approximately  $(11D \times 2D \times 2D)$  (where D is the rotor diameter), with a cell size of 2.5 m, approximately 0.0125D. This configuration results in approximately 22.5 million cells and an equal number of vortex particles. Relatively high temporal and spatial resolutions are used to ensure the accurate modeling of sweep effects. For each simulation, a time step of 0.03 s is chosen, corresponding to an azimuthal discretization of approximately  $1.5^{\circ}$ , or 240 time steps per rotor revolution. Simulations are conducted for 20,000 time steps, corresponding to 600 s of total simulation time or approximately 83 rotor revolutions. A maximum of 50 sub-iterations per time step is applied to ensure the angle of attack residual converges below a specified tolerance. Each blade is discretized radially into 50 spanwise sections using cosine spacing. The airfoil polars are obtained from fully turbulent 2-D RANS CFD simulations (Bortolotti et al., 2019).

<sup>&</sup>lt;sup>6</sup>A parameter study has been performed and showed that the results converged if the filament wake size is 1.5 revolutions or larger.

# 7.3 Blades used for the comparison

The blades used for comparison are based on the IEA-10.0-198 10 MW reference wind turbine (RWT) (Bortolotti et al., 2019), following previous studies (Li et al., 2018, 2022b, d, 2025b, 2024). The baseline straight blade is modified by aligning the half-chord line into a straight main axis. Swept blades are also used for comparison, which are modified based on the baseline straight blade by introducing *x*-components into the main-axis geometry. Afterwards, the main axis geometry is scaled so that the radius of each blade section remains unchanged compared to the baseline straight blade. The chord and twist distributions of the swept blades are also modified so that the BEM method predicts the same operational conditions (e.g., angle of attack and local thrust coefficient) as the baseline straight blade (Li et al., 2025b). The blades used for comparison include the backward swept blade mB-1 and the forward swept blade mB-5, which are also used in previous work (Li et al., 2025b). Their main axis geometries are illustrated in Fig. 4.

**Figure 4.** Front view of the backward swept blade mB-1 and the forward swept blade mB-5 in the blade root coordinate system (B-sys). The main axis, which is the half-chord line, is highlighted. The figure is adapted from (Li et al., 2025b).

It has been tested that other swept blades with different main axis shapes exhibit similar behaviors to the swept blades used in the present work. Including these additional results will not change the conclusions of the present work. The results of other swept blades are summarized in an internet appendix (Li et al., 2025a).

For all cases, the rotor radius is 99 m with a hub radius of 2.8 m. The blades are assumed to be stiff, excluding elastic deformation effects. For the low- to mid-fidelity engineering aerodynamic models, including BEM and coupled near and far wake models, the same airfoil data in the LL method is used, which is from 2-D fully turbulent RANS CFD simulations (Bortolotti et al., 2019). For BEM and coupled models, each blade is discretized radially into 80 sections.

## 7.4 Operational conditions

The operational conditions for the numerical tests also align with previous studies (Li et al., 2022b, c, d, 2025b). First, an optimal operational condition with high rotor thrust is used for comparison. The rotors operate under a uniform inflow of  $8 \text{ m s}^{-1}$  perpendicular to the rotor plane, with a constant rotational speed of  $0.855 \text{ rad s}^{-1}$  and no blade pitch.

In addition, comparisons are conducted at operational conditions corresponding to lower thrust coefficients. Three conditions defined in the IEA Wind TCP Task 37 report (Bortolotti et al., 2019) are used for these lower-loading cases, with rotational speed set at  $0.909 \,\mathrm{rad}\,\mathrm{s}^{-1}$ , wind speed varying from  $12.0 \,\mathrm{m}\,\mathrm{s}^{-1}$  to  $20.0 \,\mathrm{m}\,\mathrm{s}^{-1}$  and the blade is pitched towards lower loadings. For the baseline straight blade, the blade is pitched with  $\theta_p$  for these lower loading conditions. For swept blades, the blade is

not directly pitched but has modified chord and twist distributions according to (Li et al., 2025b), so that the BEM method predicts the same circulation distribution as the pitched straight blade.

The operational conditions are summarized in Table 2, detailing wind speed, tip-speed-ratio and pitch angle  $\theta_p$ . The table also lists the thrust and power coefficients of the rotor with baseline straight blades, as predicted by the CFD solver.

**Table 2.** Operational conditions used in the comparison.  $C_{T,\text{CFD}}^{\text{str}}$  and  $C_{P,\text{CFD}}^{\text{str}}$  are the thrust and power coefficients predicted by the CFD solver for the baseline straight blade.

| Wind speed $U_0 \text{ [m s}^{-1}\text{]}$ | Tip-speed-ratio $\lambda$ [-] | Pitch angle $\theta_p$ [ $^{\circ}$ ] | $C_{T,\mathrm{CFD}}^{\mathrm{str}}$ [-] | $C_{P,\mathrm{CFD}}^{\mathrm{str}}$ [-] |
|--------------------------------------------|-------------------------------|---------------------------------------|-----------------------------------------|-----------------------------------------|
| 8.0                                        | 10.58                         | 0.00                                  | 0.91                                    | 0.45                                    |
| 12.0                                       | 7.50                          | 5.98                                  | 0.44                                    | 0.31                                    |
| 15.0                                       | 6.00                          | 11.77                                 | 0.21                                    | 0.16                                    |
| 20.0                                       | 4.50                          | 18.51                                 | 0.09                                    | 0.07                                    |

## 585 7.5 Loads for comparison

In the present work, the non-dimensional axial and tangential loads, which are in the y- and x-directions in the blade local coordinate system (BL-sys) as described in (Li et al., 2025b), are used for comparison. The axial and tangential forces are non-dimensionalized into the local thrust coefficient and the simplified local power coefficient as follows:

$$C_t = \frac{N_B f_y^{\text{BL}} \, \mathrm{d}s}{\frac{1}{2} \rho U_0^2 2\pi r \, \mathrm{d}r},\tag{68}$$

$$\tilde{C}_p \equiv \frac{\Omega r N_B f_x^{\text{BL}} \, \mathrm{d}s}{\frac{1}{2} \rho U_0^3 2\pi r \, \mathrm{d}r},\tag{69}$$

where the simplified power coefficient  $\tilde{C}_p$  represents the contribution of the tangential force to power. Note that other loads, such as sectional moments, can also contribute to aerodynamic power. However, for moderately swept blades, these contributions are generally insignificant.

To better illustrate the influence of curved blade geometry on loads, the load differences between the swept blades and the baseline straight blade at the same radial positions are used for the comparison:

$$\Delta C_t(r) = C_t(r) - C_t^{\text{str}}(r),\tag{70}$$

$$\Delta \tilde{C}_p(r) = \tilde{C}_p(r) - \tilde{C}_p^{\text{str}}(r). \tag{71}$$

## 8 Results

In this section, comparative studies are conducted to evaluate the performance of different coupling factors and far wake models for both straight and swept blades. Building on the methodology and coupling factor modifications introduced in Sect. 6.3, the

coupled models NW-MT and NW-VC with coupling factors calculated using different methods are compared for the baseline straight blade (Str), the backward swept blade (mB-1) and the forward swept blade (mB-5). The coupled models using the original coupling factor and the two modified coupling factors are labeled as (origin), (a) and (ka), respectively. Additionally, the comparison includes coupled models using a fixed coupling factor for the swept blades, set equal to that of the baseline straight blade, labeled as (fixed). This fixed coupling factor is following the assumption that the sweep effects are fully captured in the near wake, making the coupling factor independent of blade sweep. Although insightful for comparison purposes, this approach is impractical for implementation since it requires simultaneous computations for both straight and swept blades. Finally, results from different coupled models with their preferred coupling factors are compared with BEM, LL and CFD results.

# 610 8.1 Baseline straight blade

First, loads of the baseline straight blade, calculated using the coupled models NW-MT(origin) and NW-VC(origin) with the original coupling factors, are compared with results from the BEM method, the LL solver and the RANS CFD solver at the optimal operational condition with a wind speed of  $8 \text{ m s}^{-1}$ . The results are depicted in Fig. 5.

Figure 5. Thrust coefficient  $C_t$  (panel a) and simplified power coefficient  $\tilde{C}_p$  (panel b) of the baseline straight blade Str at a wind speed of 8 m s<sup>-1</sup>, calculated using the BEM method, coupled models NW-MT(origin) and NW-VC(origin) with the original coupling factors, the LL solver and the CFD solver.

For loads in the blade root region with a radius of less than 20 m, there is a relatively large difference between results from the CFD solver and the other models. As discussed in Sect. 7, it is due to unsteady flow separation in this region. Therefore, the comparison focuses on the radius from 20 m to the blade tip.

The thrust coefficients predicted by these different models show good agreement, indicating that the predicted circulation distributions are similar (Li et al., 2025b). However, larger discrepancies are observed in the simplified power coefficient results, especially for radii larger than 60 m. All engineering aerodynamic models overestimate the tangential loads compared

to the LL and CFD results. This is likely to be related to the assumptions in the engineering models, such as the use of Prandtl's tip-loss correction in the BEM method and the assumption of a non-expanding helical trailed wake in the coupled models. Furthermore, the BEM method and the coupled models do not take into account the wake expansion and wake roll-up effects, which are significant for this high-loading condition. In contrast, the free-wake LL method includes the wake expansion and wake roll-up effects; the CFD solver is modeling the blade tip and wake effects with much higher fidelity, as the flow around the 3-D blade geometry is fully resolved.

For NW-VC(origin), there is surprisingly good agreement with the BEM results from a radius of 20 m to 90 m, for both thrust and power coefficients. Beyond 90 m, differences in the power coefficient arise due to different tip-loss modeling approaches. In comparison, NW-MT(origin) shows larger discrepancies compared to the BEM and NW-VC(origin) results for both thrust and power coefficients. Since both NW-MT(origin) and NW-VC(origin) use the same near wake model, the differences arise from the far wake modeling. Specifically, NW-MT(origin) uses the far wake momentum theory, which is less physically consistent than the far wake vortex cylinder model used in NW-VC(origin).

For operational conditions with higher wind speeds and lower thrust coefficients listed in Table 2, the results are presented in Fig. C1 in Appendix C. In general, the different models show good agreement across all tested lower loading conditions. As wind speed increases and rotor loading decreases, the agreement between different models generally improve, especially near the blade tip. This improvement is likely due to the weaker tip vortex strength and reduced induction effects at lower rotor loading conditions. Furthermore, the wake expansion effect becomes weaker as rotor loading decreases.

## 8.2 Impact of blade sweep on aerodynamic loads

In this section, the CFD results are used to illustrate the impact of blade sweep on aerodynamic loads at different operational conditions. The purpose is to provide an intuitive understanding of the relative magnitude of the blade sweep effects compared to the overall loads, setting a basis for the following comparisons. First, the CFD results of the baseline straight blade (Str) and the sweept blades (mB-1 with backward sweep and mB-5 with forward sweep) at an optimal operational condition with a wind speed of 8 m s<sup>-1</sup> are compared in Fig. 6. The relative differences in rotor-integrated thrust and power for the swept blades compared to the baseline straight blade are summarized in Table 3

Table 3. Relative differences in rotor thrust and power coefficients of the swept blades (mB-1 with backward sweep and mB-5 with forward sweep) compared to the baseline straight blade, predicted by the CFD solver. Relative differences computed as:  $\epsilon C_i = (C_i^{\text{Sweep}} - C_i^{\text{Str}})/C_i^{\text{Str}}$ , where  $C_i$  represents either the thrust coefficient  $(C_T)$  or the power coefficient  $(C_P)$ .

| Wind speed $U_0$ [m s <sup>-1</sup> ] | $\epsilon C_T^{\mathrm{mB-1}}$ [%] | $\epsilon C_T^{	ext{mB-5}}$ [%] | $\epsilon C_P^{\mathrm{mB-1}}$ [%] | $\epsilon C_P^{	ext{mB-5}}$ [%] |
|---------------------------------------|------------------------------------|---------------------------------|------------------------------------|---------------------------------|
| 8.0                                   | -0.10                              | 0.12                            | 2.90                               | -2.68                           |
| 12.0                                  | -0.02                              | 0.31                            | 0.76                               | -0.26                           |
| 15.0                                  | 0.23                               | 0.03                            | 0.82                               | -0.52                           |
| 20.0                                  | 0.17                               | -0.68                           | 0.98                               | -1.25                           |

Figure 6. Thrust coefficient  $C_t$  (panel a) and simplified power coefficient  $\tilde{C}_p$  (panel b) of the baseline straight blade Str, the backward swept blade mB-1, and the forward swept blade mB-5 at a wind speed of 8 m s<sup>-1</sup>, calculated using the CFD solver.

At this operational condition, blade sweep has a smaller relative influence on thrust than on power. For both swept blades mB-1 and mB-5, the thrust coefficients remain similar to that of the straight blade. The rotor-integrated thrust coefficient differ only by approximately 0.1%. However, significant differences are observed in the power coefficients of the swept blades compared to the straight blade, with a more substantial difference exceeding 2.5% observed in the rotor-integrated power coefficient. For radii less than 50 m, loads of the swept blades are almost identical to those of the baseline straight blade. Moving from the blade mid-span (radius of 50 m) towards the blade tip, a spanwise redistribution of loads is observed for both thrust and power coefficients, consistent with observations in previous studies (Li et al., 2018, 2022d, 2025b). Specifically, for the backward swept blade mB-1, the loads initially decrease and then increase compared to the baseline straight blade. For the forward swept blade mB-5, the load redistribution shows an opposite trend, with the loads initially increasing and then decreasing relative to the baseline straight blade.

For lower loading conditions at wind speeds of 12, 15 and 20 m s<sup>-1</sup>, the results are presented in Fig. C2 in Appendix C. Across all tested lower loading conditions, the spanwise load redistribution effect is observed for both thrust and power coefficients. However, as wind speed increases and rotor thrust coefficient decreases, the differences in loads between the swept blades and the straight blade diminish, indicating a reduced influence of blade sweep. At the highest tested wind speed of  $20 \text{ m s}^{-1}$ , corresponding to a rotor-averaged thrust coefficient of 0.1, both axial and tangential loads of the swept blades are nearly identical to those of the baseline straight blade. Nevertheless, the influence of blade sweep remains consistently more pronounced on rotor power coefficients than on rotor thrust coefficients across all tested conditions<sup>7</sup>, except for the single case of blade mB-5 operating at 12 m s<sup>-1</sup>.

<sup>&</sup>lt;sup>7</sup>Note that, in practice, for a pitch-regulated turbine operating at higher wind speeds, the pitch angle will be determined by the controller to maintain the rated aerodynamic power. Then, the differences between the swept and straight blades will be in the thrust only.

# 8.3 Coupled models with far wake momentum theory

This section evaluates the performance of different coupling factors when applied to the coupled models with far wake momentum theory. As described in Sect. 5.3, the only difference between NW-MT and NW-MT-VC is the inclusion of radial induction, which has negligible influence on the coupling factor; therefore, only NW-MT is used for this comparison. First, the rotor-averaged coupling factors  $k_{\rm FW}^{\rm MT}$  for the straight and swept blades, calculated using the three different methods, are compared in Table 4.

**Table 4.** Rotor-averaged coupling factor of the baseline straight blade (Str), the backward swept blade (mB-1) and the forward swept blade (mB-5) calculated from the coupled NW-MT model with the original coupling factor, NW-MT(origin), and the modified coupling factors, NW-MT(a) and NW-MT(ka), at a wind speed of 8 m s<sup>-1</sup>. The differences in the coupling factor of the swept blades compared to the baseline straight blade are also shown.

| Model         | Str   | mB-1  | mB-5  | $\Delta$ mB-1 | $\Delta$ mB-5 |
|---------------|-------|-------|-------|---------------|---------------|
| NW-MT(origin) | 0.904 | 0.909 | 0.838 | 0.006         | -0.066        |
| NW-MT(a)      | 0.911 | 0.912 | 0.888 | 0.001         | -0.024        |
| NW-MT(ka)     | 0.912 | 0.911 | 0.900 | -0.001        | -0.012        |

For the baseline straight blade (Str), the rotor-averaged coupling factors  $k_{\text{FW}}$  from all three methods are very similar, indicating negligible differences in far wake inductions and consequently in the loads. Thus, NW-MT(a) and NW-MT(ka) with the two modified coupling factors described in Sect. 6.3 predict very similar loads to NW-MT(origin) for the straight blade case, as shown in Fig. 5.

For the swept blades, the performance of different coupling factors is evaluated by comparing the loads predicted by the coupled models with CFD results and with NW-MT(fixed), which uses the same fixed coupling factor as the baseline straight blade. As shown in Sect. 8.2, the load differences between the swept blades and the baseline straight blade are relatively small compared to the overall loads, especially at lower loading conditions. To better illustrate the effects of using different coupling factors, the load differences between the swept blades and the baseline straight blade calculated using Eqs. (70) and (71) are compared.

At the optimal operational condition with a wind speed of 8 m s<sup>-1</sup>, the load offsets of the swept blades mB-1 and mB-5 relative to the baseline straight blade (Str) are depicted in Figs. 7 and 8, respectively. In addition, the BEM results are included as a reference, which predict approximately the same loads for the swept and straight blades (Li et al., 2025b).

For the backward swept blade mB-1, the loads predicted by the NW-MT model with different coupling factors are similar and agree well with the CFD results. Both modified methods, NW-MT(a) and NW-MT(ka), show slight improvements over NW-MT(origin) when compared to NW-MT(fixed). Furthermore, as shown in Table 4, the rotor-averaged coupling factors  $k_{\text{FW}}^{\text{MT}}$  for mB-1 are very close to those of the straight blade across all methods. For the forward swept blade mB-5, NW-

Figure 7. Offset in thrust coefficient  $(\Delta C_t)$  (panel a) and simplified power coefficient  $(\Delta \tilde{C}_p)$  (panel b) of the backward swept blade mB-1 compared to the baseline straight blade Str, at a wind speed of 8 m s<sup>-1</sup>. Results from the coupled NW-MT model with different coupling factors are compared with BEM and CFD results.

Figure 8. Offset in thrust coefficient  $(\Delta C_t)$  (panel a) and simplified power coefficient  $(\Delta \tilde{C}_p)$  (panel b) of the forward swept blade mB-5 compared to the baseline straight blade Str, at a wind speed of 8 m s<sup>-1</sup>. Results from the coupled NW-MT model with different coupling factors are compared with BEM and CFD results.

MT(origin) significantly overestimates the loads along the span. This overestimation is associated with a noticeable decrease in  $k_{\rm FW}^{\rm MT}$  compared to the straight blade, due to inconsistencies in calculating the coupling factor, as detailed in Sect. 6.2.2. In contrast, both NW-MT(a) and NW-MT(ka) with modified coupling factors show clear improvements over NW-MT(origin). However, both modified methods still slightly underestimate the rotor-averaged coupling factor  $k_{\rm FW}^{\rm MT}$ , which leads to an overall overestimation of the aerodynamic loads compared to NW-MT(fixed). Among the two methods, NW-MT(ka) shows further improvement over NW-MT(a) and has sufficiently good agreement with both NW-MT(fixed) and CFD results.

## 8.3.1 Lower loading conditions

To ensure a comprehensive evaluation, the performance of the different coupling factors applied to NW-MT is also assessed under lower loading conditions listed in Table 2. Detailed results for these conditions are shown in Figs. C3 and C4 in Appendix C, with conclusions summarized here.

For the backward swept blade mB-1, the NW-MT model with different coupling factors exhibits sufficiently good agreement with the CFD results across all tested lower loading conditions. In the case of NW-MT(origin), the loads are underestimated compared to NW-MT(fixed) from radii 20 m to 50 m and this underestimation becomes more pronounced at lower loading conditions<sup>8</sup>. Both modified methods, NW-MT(a) and NW-MT(ka), show improved agreement with NW-MT(fixed), with NW-MT(ka) showing slightly better performance. For the forward swept blade mB-5, NW-MT(origin) overestimates the loads across all conditions, showing significant discrepancies compared to the CFD results. In contrast, NW-MT(fixed) demonstrates very good agreement with the CFD solver. Results from NW-MT(a) show clear improvements over NW-MT(origin) but still overestimate the loads, especially from radii 20 m to 50 m. NW-MT(ka) shows further improvement over NW-MT(a) and has sufficiently good agreement with NW-MT(fixed).

#### 705 **8.3.2** Summary

In summary, both NW-MT(a) and NW-MT(ka) with modified coupling factors show similarly good performance as NW-MT(origin) when modeling the baseline straight blade (Str) and the backward swept blade (mB-1) across all tested operational conditions. For the forward swept blade (mB-5), NW-MT(a) offers improvement over NW-MT(origin) but still overestimates the loads. In comparison, NW-MT(ka) shows sufficiently good agreement with NW-MT(fixed) and CFD results across all tested operational conditions. Therefore, despite its more complex implementation, the modified coupling factor (ka) is preferred for coupled models using the far wake momentum theory. This suggests that NW-MT(ka) and NW-MT(ka)-VC are preferable over NW-MT and NW-MT-VC with other coupling factors.

## 8.4 Coupled models with far wake vortex cylinders

This section evaluates the performance of different coupling factors when applied to the coupled models with the far wake vortex cylinder model. As in Sect. 8.3, the influence of the modified coupling factors on both straight and swept blades is

<sup>&</sup>lt;sup>8</sup>This can be linked to the discussion in Sect. 6.2.1 that the local coupling factor  $\kappa_{\rm FW}$  maintains a lower overall level under these operational conditions.

assessed. Since the primary difference between NW-VC and NW(ideal)-VC lies in the radial induction, which has negligible influence on the coupling factor, only NW-VC is used for the comparison. The rotor-averaged coupling factors ( $k_{\rm FW}^{\rm VC}$ ) for the straight and swept blades, calculated using the three different methods, are compared in Table 5.

**Table 5.** Rotor-averaged coupling factor of the baseline straight blade (Str), the backward swept blade (mB-1) and the forward swept blade (mB-5) calculated from the original method NW-VC(origin) and the modified methods NW-VC(a) and NW-VC(ka), at a wind speed of  $8 \text{ m s}^{-1}$ . The differences in the coupling factor of the swept blades compared to the baseline straight blade are also shown.

| Model         | Str   | mB-1  | mB-5  | $\Delta$ mB-1 | $\Delta$ mB-5 |
|---------------|-------|-------|-------|---------------|---------------|
| NW-VC(origin) | 0.562 | 0.581 | 0.573 | 0.019         | 0.011         |
| NW-VC(a)      | 0.576 | 0.578 | 0.608 | 0.002         | 0.032         |
| NW-VC(ka)     | 0.576 | 0.592 | 0.583 | 0.016         | 0.006         |

For the baseline straight blade Str, the rotor-averaged coupling factors  $k_{\rm FW}$  from all three methods are similar, showing that the performance remains similar before and after the coupling factor modifications. Consequently, NW-VC(a) and NW-VC(ka) with modified coupling factors predict very similar loads as NW-VC(origin) for the straight blade case, as shown earlier in Fig. 5.

To evaluate the performance of different coupling factors for swept blades, loads predicted by the coupled models are compared with results from CFD and NW-VC(fixed), which uses a fixed coupling factor equal to that of the baseline straight blade. As in previous sections, the load differences between the swept blades and the baseline straight blade, calculated using Eqs. (70) and (71) are compared, to better illustrate the effects of using different coupling factors. At the optimal operational condition with a wind speed of 8 m s<sup>-1</sup>, the load offsets between the swept blades mB-1 and mB-5 relative to the baseline straight blade Str are shown in Figs. 9 and 10, respectively. The BEM results are also included as a reference, which predict approximately the same loads for the swept and straight blades (Li et al., 2025b).

For both swept blades mB-1 and mB-5, loads from the coupled models with different coupling factors are very similar to each other and show good agreement with the CFD results. As shown in Table 5, the rotor-averaged coupling factors  $k_{\rm FW}^{\rm VC}$  for both swept blades from different methods are all similar and closely approximate the value of the straight blade Str.

Compared to NW-MT(origin), the results from NW-VC(origin) show significant improvements in modeling the forward swept blade mB-5, despite using the original coupling factor. To understand the surprisingly good performance of the NW-VC(origin) model, the analysis is performed using the similar method as in Sect. 6.2 for NW-MT(origin). Specifically, the local coupling factor and also the different components of the axial inductions for the swept blades mB-1 and mB-5 are compared with those of the baseline straight blade Str, as shown in Fig. 11.

For the baseline straight blade Str, the axial induction difference  $\delta a$  is approximately zero from the mid-span to a radius of 90 m. Beyond 90 m,  $\delta a$  decreases steeply. This indicates that the axial induction from NW-VC(origin) is almost identical

Figure 9. Offset in thrust coefficient  $(\Delta C_t)$  (panel a) and simplified power coefficient  $(\Delta \tilde{C}_p)$  (panel b) of the backward swept blade mB-1 compared to the baseline straight blade Str, at a wind speed of 8 m s<sup>-1</sup>. The results calculated from the coupled NW-VC model with different coupling factors are compared with BEM and CFD results.

Figure 10. Offset in thrust coefficient  $(\Delta C_t)$  (panel a) and simplified power coefficient  $(\Delta \tilde{C}_p)$  (panel b) of the forward swept blade mB-5 compared to the baseline straight blade Str, at a wind speed of 8 m s<sup>-1</sup>. The results calculated from the coupled NW-VC model with different coupling factors are compared with BEM and CFD results.

Figure 11. Comparison of: (a) the local coupling factor  $\tilde{\kappa}_{FW}$ ; (b) different components of the axial induction factors from the coupled model NW-VC(origin) with the existing coupling factor, of the baseline straight blade Str, the backward swept blade mB-1 and the forward swept blade mB-5, at a wind speed of 8 m s<sup>-1</sup>.

to the BEM axial induction along the blade span, except in the blade tip region with radius larger than 90 m. In contrast, as shown earlier in Fig. 3, NW-MT(origin) that has the same method of calculating the coupling factor has noticeable non-zero values of  $\delta a$  from a radius of 20 m to 90 m. This means that BEM and NW-MT have larger discrepancies in axial inductions along the span. This observation further justifies the far wake vortex cylinder model in NW-VC provides a more physically consistent representation of the far wake compared to the simple scaling approach used in the far wake momentum theory of NW-MT. For the straight blade Str, the improved consistency reduces systematic errors in axial induction along the span, making the coupling between the near and far wake models inherently easier to balance. For the swept blades mB-1 and mB-5, although the blade sweep effects make balancing of  $\delta a$  more challenging, the improved physical consistency of the far wake vortex cylinder model allows for effective balancing, even when using the original coupling factor. Additionally, the NW-VC model has a different definition of the coupling factor compared to that of NW-MT, as described in Sect. 4.2.1, which prevents extraordinary changes in the far wake induction. Furthermore, variations in the rotor-averaged coupling factor  $k_{\rm FW}^{\rm VC}$  affect the far wake inductions differently in the far wake vortex cylinder model than  $k_{\rm FW}^{\rm MT}$  in the simple scaling approach used in the far wake momentum theory.

# 8.4.1 Lower loading conditions

To provide a comprehensive evaluation, the performance of the different coupling factors applied to NW-VC is also assessed under lower loading conditions listed in Table 2. Detailed results for these conditions are provided in Figs. C5 and C6 in Appendix C, with conclusions summarized here.

For the backward swept blade mB-1, results from NW-VC with different coupling factors exhibit very good agreement with the CFD results, particularly from the mid-span to the blade tip. Overall, results from NW-VC(origin) show sufficiently good

agreement with NW-VC(fixed) and CFD results. Some minor differences are observed from radius 20 m to 50 m, especially for NW-VC(origin) at a wind speed of 20 m s<sup>-1</sup>. Both modified methods, NW-VC(a) and NW-VC(ka), show marginally improved agreement with NW-VC(fixed), especially from radius 20 m to 50 m.

For the forward swept blade mB-5, NW-VC(origin) results maintain sufficiently good agreement with NW-VC(fixed) and CFD results at wind speeds of 12 m s<sup>-1</sup> and 15 m s<sup>-1</sup>. However, at a wind speed of 20 m s<sup>-1</sup>, corresponding to a rotor thrust coefficient of 0.1, NW-VC(origin) significantly underestimates the loads between radii of 20 m and 70 m. Nevertheless, results from NW-VC(origin) still show significantly improved agreement compared to NW-MT(origin), due to the more physically consistent far wake vortex cylinder model. The two modified methods, NW-VC(a) and NW-VC(ka), demonstrate further improvements over NW-VC(origin) and provide better agreement with NW-VC(fixed). At 20 m s<sup>-1</sup>, NW-VC(a) shows improvements over NW-VC(origin) but noticeably overestimates the load between radii of 20 m and 50 m. In comparison, NW-VC(ka) shows further improved agreement with NW-VC(fixed) compared to NW-VC(a). However, at such low loading conditions, the magnitude of the absolute error is very small, so the performance of both NW-VC(a) and NW-VC(ka) remains acceptable.

# **8.4.2** Summary

In summary, NW-VC(origin) with the original coupling factor shows good agreement with NW-VC(fixed) and CFD results at the optimal operational condition but its performance worsens under lower loading conditions. Nevertheless, due to its more physically consistent far wake modeling, NW-VC(origin) performs significantly better than NW-MT(origin), especially for forward swept blades. Both NW-VC(a) and NW-VC(ka) with modified coupling factors show further improvements over NW-VC(origin) and provide sufficiently good performance across different operational conditions. However, the modified method (a) is preferred for coupled models using the far wake vortex cylinder model due to its simpler implementation. This suggests that NW-VC(a) and NW(ideal)-VC(a) should be used over NW-VC and NW(ideal)-VC with other coupling factors.

# 8.5 Comparison of different coupled methods

In this section, the performance of different coupled methods is compared with LL and CFD results. The methods evaluated include the existing coupled model NW-MT(ka) and the modified coupled models NW-MT(ka)-VC, NW-VC(a) and NW(ideal)-VC(a). Note that all coupled models use the favorable modified methods for calculating the coupling factor introduced earlier. Results from the conventional BEM method are also included as a baseline. The primary aim is to compare the performance of these coupled models and investigate the impact of different trailed wake radial induction modeling and far wake modeling on the load calculation of swept blades.

At the optimal operational condition with a wind speed of 8 m s<sup>-1</sup>, the load offsets of the swept blades mB-1 and mB-5 compared to the baseline straight blade are shown in Figs. 12 and 13, respectively.

The BEM method predicts that the swept blades have approximately the same loads as the baseline straight blade, which is expected since it is not able to model the influence of blade sweep on the wake. For the thrust coefficient, the existing coupled model NW-MT(ka), which does not include trailed wake radial induction, predicts results that differ from all the

Figure 12. Offset in thrust coefficient  $(\Delta C_t)$  (panel a) and simplified power coefficient  $(\Delta \tilde{C}_p)$  (panel b) of the backward swept blade mB-1 compared to the baseline straight blade Str, at a wind speed of 8 m s<sup>-1</sup>. Results calculated from NW-MT(ka), NW-MT(ka)-VC, NW-VC(a) and NW(ideal)-VC(a) are compared with BEM, LL and CFD results.

Figure 13. Offset in thrust coefficient  $(\Delta C_t)$  (panel a) and simplified power coefficient  $(\Delta \tilde{C}_p)$  (panel b) of the forward swept blade mB-5 compared to the baseline straight blade Str, at a wind speed of 8 m s<sup>-1</sup>. Results calculated from NW-MT(ka), NW-MT(ka)-VC, NW-VC(a) and NW(ideal)-VC(a) are compared with BEM, LL and CFD results.

modified coupled models that include the trailed wake radial induction. This difference is relatively small compared to the overall magnitude of the thrust coefficient. Nevertheless, including the trailed wake radial induction, as done in the modified coupled models, leads to predicted thrust coefficients showing improved agreement with the LL and CFD results. Furthermore, comparing the results from NW-VC(a) and NW(ideal)-VC(a), the differences are negligible, indicating that the influence of the finite number of blades on radial induction is negligible for swept blades. In addition, the good agreement also relates to the high accuracy of the approximated near wake axial and tangential inductions in the NW-VC model, as described in Sect. 3.2. For the simplified power coefficient, results from NW-MT(ka) are almost identical to those from NW-MT(ka)-VC, indicating that the radial induction has negligible contribution to tangential loads for blades with only sweep. In general, the existing coupled model NW-MT(ka) with the modified coupling factor demonstrates sufficiently good performance for both swept blades. Moreover, NW-VC(a) and NW(ideal)-VC(a) show very good agreement with LL and CFD results, with only insignificant improvements compared to NW-MT(ka) and NW-MT(ka)-VC. The slight differences are mainly due to the far wake modeling, with a small portion possibly resulting from the different coupling factors applied.

Furthermore, the relative differences in rotor-integrated thrust and power coefficients for the swept blades compared to the baseline straight blade are summarized in Table 6.

**Table 6.** Relative differences in rotor-integrated thrust and power coefficients of the swept blades mB-1 (backward swept) and mB-5 (forward swept) compared to the baseline straight blade, at a wind speed of 8 m s<sup>-1</sup>. Results from coupled methods NW-MT(ka), NW-MT(ka)-VC, NW-VC(a) and NW(ideal)-VC(a) are compared against BEM, LL and CFD results.

| Method          | $\epsilon C_T^{	ext{mB-1}}$ [%] | $\epsilon C_T^{	ext{mB-5}}$ [%] | $\epsilon C_P^{\mathrm{mB-1}}$ [%] | $\epsilon C_P^{\mathrm{mB-5}}$ [%] |
|-----------------|---------------------------------|---------------------------------|------------------------------------|------------------------------------|
| BEM             | 0.01                            | 0.01                            | 0.25                               | 0.25                               |
| NW-MT(ka)       | 0.26                            | 0.11                            | 2.15                               | -0.45                              |
| NW-MT(ka)-VC    | -0.33                           | 0.72                            | 2.22                               | -0.52                              |
| NW-VC(a)        | -0.21                           | 0.45                            | 2.67                               | -1.31                              |
| NW(ideal)-VC(a) | -0.12                           | 0.37                            | 2.67                               | -1.30                              |
| LL              | -0.20                           | 0.06                            | 1.90                               | -1.54                              |
| CFD             | -0.10                           | 0.12                            | 2.90                               | -2.68                              |

As presented in Table 6, all coupled models predict relatively small differences in rotor thrust coefficients compared to the baseline straight blade. However, rotor power coefficients exhibit larger relative differences. Among the coupled methods, NW-VC(a) and NW(ideal)-VC(a) exhibit the closest agreement with LL and CFD predictions, demonstrating improved performance over NW-MT(ka) and NW-MT(ka)-VC. These findings highlight the effectiveness of the modified coupled models, particularly NW-VC(a), in accurately capturing the sweep-induced effects on rotor-integrated loads. In contrast, the conventional BEM method predicts negligible differences in both thrust and power, as expected, since it does not model wake-induced effects of blade sweep.

# 8.5.1 Lower loading conditions

15 For the lower loading conditions listed in Table 2, the results are depicted in Figs. C7 and C8 in Appendix C, with conclusions summarized here.

For the backward swept blade mB-1, there is very good agreement among the different coupled models, all showing very similar trends with the LL and CFD results across all tested operational conditions. Specifically, as the load decreases, the thrust coefficients from NW-MT(ka) and NW-MT(ka)-VC become very similar, indicating that the relative importance of the trailed wake radial induction decreases with the loading. For the forward swept blade mB-5, the results are almost identical for models with the same far wake modeling. For radii between 20 m and 50 m, results from models with different far wake modeling show slight differences. Note that these differences can also result from the different coupling factors applied.

At lower loading conditions, load offsets from CFD show notably difference from both the engineering models and the LL solver, around a radius of 60 m. The load offset is lower for the backward swept blades and is higher for the forward swept blades. This difference is also visualized for the tangential load of the swept blades at the optimal operational condition, as shown in Figs. 12 and 13. This difference could be related to the assumption of cross-flow principle and the use of 2-D airfoil data. The actual 3-D flow effects may introduce additional complexities and result in secondary effects, which are only captured by the CFD solver. For example, spanwise flow can lead to variations in the viscous drag force, as analyzed by Gaunaa et al. (2024). As discussed in Sects. 8.3.1 and 8.4.1, the influence of induction becomes less significant for lower loading conditions. Consequently, the absolute differences between different coupled methods are negligible and all methods predict very similar results.

## **8.5.2** Summary

In summary, all modified coupled models using their respective favorable modified coupling factors show good agreement with LL and CFD results for modeling the loads of swept blades. Notably, the modified coupled models exhibit an exceptionally close match with the free-wake LL solver, indicating the good capability in capturing sweep effects. The computationally efficient simplified coupled models NW-MT(ka)-VC and NW-VC(a) have similarly good performance as the idealized NW(ideal)-VC(a) model for modeling sweep effects. It is also shown that modeling the trailed wake radial induction is beneficial for predicting the thrust coefficient of swept blades, while the influence of a finite number of blades on the radial induction are less significant.

## 840 8.6 Computational effort

This section first reports the computational effort required to obtain the steady-state results presented in this study <sup>9</sup>, followed by a summary of indicative CPU times for unsteady time-marching aeroelastic simulations.

<sup>&</sup>lt;sup>9</sup>The CPU time for post-processing is included in this comparison and remains constant across all methods. Additionally, all presented results are converged to steady-state solutions. Therefore, the computational time reported here is significantly longer than the computational time needed per time step in a time-marching simulation.

For the stand-alone BEM method, a single steady-state computation takes approximately 2 s on one CPU core. Coupled methods using the approximated near wake induction (NW-MT, NW-MT-VC and NW-VC) require about 6 s per steady-state computation. In contrast, the idealized coupled model NW(ideal)-VC needs significantly more computational effort, taking approximately 180 s. The higher-fidelity LL and CFD solvers require substantially greater computational resources. Both solvers are parallelized using the Message Passing Interface (MPI) and executed on the Sophia HPC cluster, where each node consists of 32 CPU cores operating at 2.9 GHz. Specifically, the LL solver uses 128 cores per computation with a wall-time of approximately 26.5 h. The CFD solver utilizes 216 cores per computation, with a wall-time of approximately 1.2 h.

For unsteady time-marching aeroelastic simulations, the computational effort depends on the simulation setup details of both the aerodynamic and structural solvers. An early version of the NW-MT-VC model has been implemented in the aeroelastic solver HAWC2 (version 13.0.0), with computational effort similar to the NW-MT model; see Section 8.8 of Li et al. (2022d) for a more detailed discussion of aeroelastic simulation times using BEM and NW-MT. For a typical time-marching aeroelastic setup, the total CPU time when using the NW-MT-VC method is approximately twice that of the BEM method. In comparison, the total CPU time for an unsteady aeroelastic simulation with LL or actuator line (AL) methods (e.g., HAWC2 coupled to the MIRAS solver or the AL module in EllipSys3D) is approximately 3-4 orders of magnitude larger<sup>10</sup>. The NW-VC model, by contrast, has not been implemented in HAWC2, since its computational effort is expected to be substantially higher. While promising in theory, its new coupling approach requires expensive elliptic integrals to be recalculated at nearly every time step as the rotor loading changes, which currently limits its practical use in aeroelastic simulations. By comparison, the NW-MT-VC method only needs to update the elliptic integrals when the blade geometry changes exceed a specified tolerance. Consequently, the NW-MT-VC method provides a favorable balance between aerodynamic fidelity and computational efficiency, making it particularly suitable for use in unsteady time-marching aeroelastic simulations.

## 9 Conclusions and future work

This study addresses the challenge of accurately and efficiently calculating aerodynamic loads on swept wind turbine blades. We introduce a novel and computationally efficient method that couples near wake bound and trailed vortex modeling with a far wake vortex cylinder model. This approach offers a more physically consistent representation of the far wake compared to previous coupled models based on the far wake momentum theory. The improved coupling method between the near and far wake models enables automatic adjustment of the coupling factor across various loading conditions and different swept blade configurations. The modified coupled models, with the proposed coupling factors, show significantly improved agreement with higher-fidelity free-wake lifting line (LL) and computational fluid dynamics (CFD) simulations, particularly for forward swept blades, where previous models showed limitations. For blades with backward sweep, the new models presented here predict both the distributed and integrated aerodynamic loads close to the LL and CFD results, while for forward swept blades, the trends of the distributed loads are well captured, although small discrepancies remain in the integrated load predictions. In comparison, the conventional blade element momentum (BEM) method is unable to capture either distributed or integrated

<sup>&</sup>lt;sup>10</sup>Note that the LL and AL methods can be accelerated using parallel computing and GPU acceleration to reduce wall-time.

load variations due to backward or forward blade sweep. Moreover, the proposed NW-MT-VC model is readily applicable to unsteady aeroelastic simulations within the existing framework, with the computational effort comparable to those based on the conventional BEM method. This makes the NW-MT-VC model highly suitable for aero-servo-elastic simulations and the design optimization of swept blades. By contrast, the NW-VC model is well suited for aerodynamic steady-state computations and design optimization, but requires further acceleration (e.g., reuse of elliptic integrals) before becoming practical for unsteady aeroelastic simulations.

Looking forward, the coupled near wake and far wake vortex cylinder model presented in this study has promising potential for further adaptation to model more generalized curved blades, including configurations with combined sweep and prebend. This presents a valuable direction for future research and development. Future work will also focus on improving the unsteady aerodynamic responses of the models, such as through a detailed investigation of the factors used in the indicial response functions. Further, systematic comparisons will be performed with unsteady LL and actuator line (AL) simulations for both unsteady aerodynamic and aeroelastic cases.

## **Appendix A: Nomenclature**

**Table A1.** Variables used in the present work.

| Symbol               | Description                                                                 |
|----------------------|-----------------------------------------------------------------------------|
| $\overline{a, a'}$   | Axial and tangential induction factors                                      |
| $C_L$                | Lift coefficient                                                            |
| $C_t$                | Local thrust coefficient                                                    |
| $C_T$                | Rotor-integrated thrust coefficient                                         |
| $\tilde{C}_p$        | Local simplified power coefficient                                          |
| $C_P$                | Rotor-integrated power coefficient                                          |
| D                    | Rotor diameter                                                              |
| F                    | Tip-loss factor                                                             |
| h                    | Helical pitch                                                               |
| k                    | Factor for the calculation of the elliptic integral                         |
| $k_1, k_2, k_3$      | Factors for the relationship between axial induction and thrust coefficient |
| $k_s$                | Normalized sectional circulation of the vortex cylinder                     |
| $N_B$                | Number of blades                                                            |
| r, R                 | Radius of the calculation point; trailing point                             |
| $u_a, u_t, u_r$      | Axial, tangential and radial induced velocities                             |
| $U_0$                | Free-stream wind speed                                                      |
| $V_{ m rel}$         | Relative velocity                                                           |
| $\Gamma, \Gamma_B$   | Bound vorticity strength of all blades; of a single blade                   |
| $\gamma_t, \gamma_l$ | Tangential and longitudinal vorticity strength of the vortex cylinder       |
| $\rho$               | Air density                                                                 |
| $\Omega$             | Rotor speed                                                                 |

## Appendix B: Partial derivative for the far wake vortex cylinder model

This section derives the partial derivative of the far wake axial induction  $a_{\text{FW,VC}}$  from the far wake vortex cylinder model with respect to the rotor-averaged coupling factor  $k_{\text{FW}}^{\text{VC}}$ . This derivative is a critical component for calculating the coupling factor in the far wake vortex cylinder model.

By substituting Eq. (38) into Eq. (43), the partial derivative is expressed as:

$$\frac{\partial a_{\text{FW,VC}}}{\partial k_{\text{FW}}^{\text{VC}}} = \frac{\bar{h}_{\text{NW}}}{4U_0} \sum_{j=1}^{N_{\text{tp}}} \frac{\partial \tilde{u}_a}{\partial y} (r, R_j, y - y_{\text{FW},j}) \gamma_{t,j}. \tag{B1}$$

The next step is to determine  $\partial \tilde{u}_a/\partial y$ . According to Eq. (30), the partial derivative is as follows:

$$\frac{\partial \tilde{u}_a}{\partial y} = \frac{\partial}{\partial y} \left[ g_c(y) \left( K(m) + \frac{R - r}{R + r} \Pi(n, m) \right) \right], \tag{B2}$$

where  $g_c(y)$  is defined as:

$$g_c(y) \equiv \frac{y\sqrt{m}}{4\pi\sqrt{rR}}.$$
(B3)

Using the chain rule, Eq. (B2) expands to:

$$\frac{\partial \tilde{u}_a}{\partial y} = \frac{\partial g_c}{\partial y} \left( K(m) + \frac{R - r}{R + r} \Pi(n, m) \right) + g_c(y) \frac{\partial m}{\partial y} \left( \frac{\mathrm{d}K(m)}{\mathrm{d}m} + \frac{R - r}{R + r} \frac{\partial \Pi(n, m)}{\partial m} \right). \tag{B4}$$

The parameter m is defined according to Eq. (33). Its partial derivative with respect to y is:

$$\frac{\partial m}{\partial y} = \frac{-2ym}{(R+r)^2 + y^2}. ag{B5}$$

Using the definition of  $g_c(y)$  in Eq. (B3) and applying the chain rule. Also, inserting Eq. (B5):

$$\frac{\partial g_c}{\partial y} = \frac{1}{4\pi\sqrt{rR}} \left( \sqrt{m} + \frac{y}{2\sqrt{m}} \frac{\partial m}{\partial y} \right) = \frac{1}{2\pi} \frac{(R+r)^2}{\left[ (R+r)^2 + y^2 \right]^{\frac{3}{2}}}.$$
 (B6)

The derivatives of the complete elliptic integrals of the first and third kinds with respect to m are given by:

$$\frac{\mathrm{d}K\left(m\right)}{\mathrm{d}m} = \frac{E(m) - (1 - m)K(m)}{2m(1 - m)},\tag{B7}$$

$$\frac{\partial \Pi(n,m)}{\partial m} = \frac{E(m) - (1-m)\Pi(n,m)}{2(m-n)(1-m)}.$$
(B8)

Finally, by substituting Eqs. (B5) to (B8) into Eq. (B4), the complete expression for the partial derivative is obtained.

**Figure C1.** Thrust coefficient  $C_t$  and simplified power coefficient  $\tilde{C}_p$  of the baseline straight blade. Values are calculated using the BEM method, the coupled methods NW-MT and NW-VC with the original coupling factor, the LL solver and the CFD solver. Results are shown for three operational conditions: (a), (b) at 12 m s<sup>-1</sup>; (c), (d) at 15 m s<sup>-1</sup>; and (e), (f) at 20 m s<sup>-1</sup>.

**Figure C2.** Thrust coefficient  $C_t$  and simplified power coefficient  $\tilde{C}_p$  of the baseline straight blade, the backward swept blade mB-1 and the forward swept blade mB-5, calculated using the CFD solver. Results are shown for three operational conditions: (a), (b) at  $12 \text{ m s}^{-1}$ ; (c), (d) at  $15 \text{ m s}^{-1}$ ; and (e), (f) at  $20 \text{ m s}^{-1}$ .

Figure C3. Offset in thrust coefficient  $(\Delta C_t)$  and simplified power coefficient  $(\Delta \tilde{C}_p)$  of the backward swept blade mB-1 compared to the baseline straight blade Str. Values are calculated using the coupled NW-MT model with different coupling factors, the BEM method and the CFD solver. Results are shown for three operational conditions: (a), (b) at 12 m s<sup>-1</sup>; (c), (d) at 15 m s<sup>-1</sup>; and (e), (f) at 20 m s<sup>-1</sup>.

Figure C4. Offset in thrust coefficient  $(\Delta C_t)$  and simplified power coefficient  $(\Delta \tilde{C}_p)$  of the forward swept blade mB-5 compared to the baseline straight blade Str. Values are calculated using the coupled NW-MT model with different coupling factors, the BEM method and the CFD solver. Results are shown for three operational conditions: (a), (b) at 12 m s<sup>-1</sup>; (c), (d) at 15 m s<sup>-1</sup>; and (e), (f) at 20 m s<sup>-1</sup>.

Figure C5. Offset in thrust coefficient  $(\Delta C_t)$  and simplified power coefficient  $(\Delta \tilde{C}_p)$  of the backward swept blade mB-1 compared to the baseline straight blade Str. Values are calculated using the coupled NW-VC model with different coupling factors, the BEM method and the CFD solver. Results are shown for three operational conditions: (a), (b) at 12 m s<sup>-1</sup>; (c), (d) at 15 m s<sup>-1</sup>; and (e), (f) at 20 m s<sup>-1</sup>.

Figure C6. Offset in thrust coefficient  $(\Delta C_t)$  and simplified power coefficient  $(\Delta \tilde{C}_p)$  of the forward swept blade mB-5 compared to the baseline straight blade Str. Values are calculated using the coupled NW-VC model with different coupling factors, the BEM method and the CFD solver. Results are shown for three operational conditions: (a), (b) at 12 m s<sup>-1</sup>; (c), (d) at 15 m s<sup>-1</sup>; and (e), (f) at 20 m s<sup>-1</sup>.

Figure C7. Offset in thrust coefficient  $(\Delta C_t)$  and simplified power coefficient  $(\Delta \tilde{C}_p)$  of the backward swept blade mB-1 compared to the baseline straight blade Str. Values are calculated using different coupled models, the BEM method, the LL solver and the CFD solver. Results are shown for three operational conditions: (a), (b) at 12 m s<sup>-1</sup>; (c), (d) at 15 m s<sup>-1</sup>; and (e), (f) at 20 m s<sup>-1</sup>.

Figure C8. Offset in thrust coefficient  $(\Delta C_t)$  and simplified power coefficient  $(\Delta \tilde{C}_p)$  of the forward swept blade mB-5 compared to the baseline straight blade Str. Values are calculated using different coupled models, the BEM method, the LL solver and the CFD solver. Results are shown for three operational conditions: (a), (b) at 12 m s<sup>-1</sup>; (c), (d) at 15 m s<sup>-1</sup>; and (e), (f) at 20 m s<sup>-1</sup>.

Data availability. The 2-D airfoil data used in this article are generated with 2-D fully turbulent RANS computations (Bortolotti et al., 2019).

Author contributions. This work builds on previous research by AL, GRP and MG. The concept of using a vortex cylinder model as the far wake model originated from MG, AL and GRP. The modified coupled near and far wake models were proposed and described by AL, with contributions from GRP and MG. The investigation into the existing method of calculating the coupling factor was led by AL, with input from MG and GRP. The modified coupling methods were proposed by AL, with input from MG and GRP. AL performed computations using the RANS CFD solver, the free-wake lifting line solver and engineering aerodynamic models. AL also performed the post-processing of the simulation results and plotting the figures. All authors jointly contributed to the conclusions of this work and to the writing of this manuscript.

Competing interests. DTU Wind and Energy Systems develops and distributes the Navier–Stokes solver EllipSys3D on commercial and academic terms. DTU Wind and Energy Systems also develops, supports and distributes HAWC2 on commercial terms, and HAWC2 is available free of charge for educational and academic research purposes.

Acknowledgements. The authors would like to thank their colleague Frederik Zahle at DTU Wind and Energy Systems for setting up the fully-scripted mesh generation and post-processing for the Reynolds-averaged Navier–Stokes (RANS) simulations in EllipSys3D. Computational and storage resources were provided by the Sophia HPC Cluster at DTU (DOI: 10.57940/FAFC-6M81). This work was supported by the AMTip project, funded by the Energy Technology Development and Demonstration Program (EUDP) (Case no. 64021-2062). This paper was revised with the assistance of AI tools, including OpenAI's ChatGPT (GPT-4, GPT-40, GPT-4.1, GPT-4.5, o1-preview and o1), which were used to generate suggestions to improve language and wording based on an existing draft. The manuscript underwent multiple rounds of revision by the authors, with AI-generated suggestions selectively incorporated and extensively modified.

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
