# Peer review of "Computationally efficient aerodynamic modeling of swept wind turbine blades using coupled near wake and vortex cylinder models"

_Wind Energy Science, 2025_

## Author Comment (AC1)

September 19, 2025

Dear Reviewers,

First of all, we would like to sincerely thank you for your constructive and detailed comments on our manuscript. We have carefully addressed all of your suggestions and concerns. Please find below our responses (in black) to your comments (in blue). In addition to the changes suggested by the reviewers, we have also made minor edits throughout the manuscript to improve clarity and readability.

For your convenience, we have included a marked-up version of the revised manuscript showing all changes.

Yours sincerely,

Ang Li, Mac Gaunaa and Georg Raimund Pirrung

Technical University of Denmark
**Department of Wind and Energy Systems**

Frederiksborgvej 399
Building 114

DK-4000 Roskilde

angl@dtu.dk
www.dtu.dk

[Figure]

**Comments to Reviewer #1**

Dear authors,

I had the pleasure to read your study and found it very well organized and complete (both in terms of theory formulation and data analysis). As far as I could check, I did not find any technical issue to be fixed. My only comment is that a little bit more discussion should be added on the switch to the unsteady formulation. Are you expecting it to perform as well as a LLT or an ALM? If it is indeed true that the compuational cost is lower and comparable to BEM, the other two methods are known to perform well in dynamic mode and can be also easily coupled with a structural solver.

We thank the reviewer for the positive feedback and for highlighting this important aspect.

In response to the comment, we have added further discussion regarding the unsteady formulation in the Introduction, Section 8.6, and Section 9. Specifically, we clarify the computational effort required for unsteady simulations and compare the performance to lifting-line (LL) and actuator line (AL) methods.

Regarding the accuracy of the coupled near- and far-wake model, we now explicitly refer to [1], which compared the aerodynamic response predicted by the NW-MT method and a free-wake lifting-line model for pitch step and prescribed vibration cases. It should be noted that the NW-MT method is equivalent to the NW-MT-VC method proposed in the present work but subtracts the vortex cylinder model. For the computational effort, we refer to [2], which provides a detailed discussion of aeroelastic simulation times using both BEM and NW-MT methods.

Through this analysis, we highlight that the proposed NW-MT-VC model provides a favorable balance between aerodynamic fidelity and computational efficiency, making it particularly suitable for use in unsteady time-marching aeroelastic simulations. In contrast, we expect the NW-VC model to have substantially higher computational cost, due to the need to recompute elliptic integrals in the vortex cylinder model during unsteady simulation.

We have also expanded the discussion on potential further improvements to the unsteady aerodynamic response of the model, as well as the need for systematic comparisons with unsteady lifting-line (LL) and actuator line (AL) simulations for both unsteady aerodynamic and aeroelastic cases.

We have added the following descriptions in the Introduction, Section 8.6, and Section 9:

In the **Introduction**: *Regarding accuracy in unsteady aerodynamic simulations, previous studies have shown that for a straight blade, the NW-MT approach can closely reproduce the aerodynamic response predicted by free-wake lifting line models, as demonstrated for pitch steps and prescribed vibration cases [1]. This represents a significant improvement over the BEM method.*

In **Section 8.6**: *For unsteady time-marching aeroelastic simulations, the computational effort depends on the simulation setup details of both the aerodynamic and structural solvers. An early version of the NW-MT-VC model has been implemented in the aeroelastic solver HAWC2 (version*

[Figure]

*13.0.0), with computational effort similar to the NW-MT model; see Section 8.8 of [2] for a more detailed discussion of aeroelastic simulation times using BEM and NW-MT. For a typical time-marching aeroelastic setup, the total CPU time when using the NW-MT-VC method is approximately twice that of the BEM method. In comparison, the total CPU time for an unsteady aeroelastic simulation with LL or actuator line (AL) methods (e.g., HAWC2 coupled to the MIRAS solver or the AL module in EllipSys3D) is approximately 3-4 orders of magnitude larger(Note that the LL and AL methods can be accelerated using parallel computing and GPU acceleration to reduce wall-time.). The NW-VC model, by contrast, has not been implemented in HAWC2, since its computational effort is expected to be substantially higher. While promising in theory, its new coupling approach requires expensive elliptic integrals to be recalculated at nearly every time step as the rotor loading changes, which currently limits its practical use in aeroelastic simulations. By comparison, the NW-MT-VC method only needs to update the elliptic integrals when the blade geometry changes exceed a specified tolerance. Consequently, the NW-MT-VC method provides a favorable balance between aerodynamic fidelity and computational efficiency, making it particularly suitable for use in unsteady time-marching aeroelastic simulations.*

In **Section 9**: *Moreover, the proposed NW-MT-VC model is readily applicable to unsteady aeroelastic simulations within the existing framework, with the computational effort comparable to those based on the conventional BEM method. This makes the NW-MT-VC model highly suitable for aero-servo-elastic simulations and the design optimization of swept blades. By contrast, the NW-VC model is well suited for aerodynamic steady-state computations and design optimization, but requires further acceleration (e.g., reuse of elliptic integrals) before becoming practical for unsteady aeroelastic simulations.*

*Future work will also focus on improving the unsteady aerodynamic responses of the models, such as through a detailed investigation of the factors used in the indicial response functions. Further, systematic comparisons will be performed with unsteady LL and actuator line (AL) simulations for both unsteady aerodynamic and aeroelastic cases.*

[Figure]

**Comments to Reviewer #2**

The study proposes a computationally efficient model that improves the load prediction for swept blades by coupling a near-wake representation with a far-wake vortex cylinder model, while modifying the coupling factor to enhance numerical stability and accuracy; validated under steady-state conditions, the model demonstrates accuracy comparable to high-fidelity methods (such as free-wake lifting-line models and RANS simulations) but with significantly lower computational cost. The overall work is innovative, methodologically rigorous, and represents a high-quality contribution in the field of wind turbine aerodynamic modeling.

Only two minor improvements suggested.

1. As Helical Angle phi is not defined nearby Eqs. (15-17) and not shown in Fig. 1, it is better to difine this variable nearby rather than below Eq. (22).

We thank the reviewer for the positive feedback and for highlighting this point. In response, we have now defined the helix angle $\varphi$ in Eq. (14). We also clarify in the manuscript that the helix angle $\varphi$ corresponds to the trailed vortex, and is calculated from the inductions at the trailing point (tp). Additionally, we have introduced the variable $\beta$ before Eq. (11) to further improve clarity.

2. In the Author contribution part: AL performed computations of using the RANS CFD solver, ...

If 'of' is deleted, the paper is more perfect.

We thank the reviewer for catching this. The sentence in the Author contributions section has been corrected as suggested.

[Figure]

**Bibliography**

[revised manuscript text omitted]
_{\mathrm{FW,VC}}}{\partial k_{\mathrm{FW}}^{\mathrm{VC}}} = \frac{\bar{h}_{\mathrm{NW}}}{4U_0} \sum_{j=1}^{N_{\mathrm{tp}}} \frac{\partial \tilde{u}_a}{\partial y}(r, R_j, y - y_{\mathrm{FW},j})\gamma_{t,j}. \tag{B1}$$

The next step is to determine $\partial \tilde{u}_a/\partial y$. According to Eq. (30), the partial derivative is as follows:

$$\frac{\partial \tilde{u}_a}{\partial y} = \frac{\partial}{\partial y}\left[ g_c(y)\left( K\left(m\right) + \frac{R-r}{R+r}\Pi\left(n,m\right) \right) \right], \tag{B2}$$

where $g_c(y)$ is defined as:

$$g_c(y) \equiv \frac{y\sqrt{m}}{4\pi\sqrt{rR}}. \tag{B3}$$

Using the chain rule, Eq. (B2) expands to:

$$\frac{\partial \tilde{u}_a}{\partial y} = \frac{\partial g_c}{\partial y}\left( K\left(m\right) + \frac{R-r}{R+r}\Pi\left(n,m\right) \right) + g_c(y)\frac{\partial m}{\partial y}\left( \frac{\mathrm{d}K\left(m\right)}{\mathrm{d}m} + \frac{R-r}{R+r}\frac{\partial \Pi\left(n,m\right)}{\partial m} \right). \tag{B4}$$

[revised manuscript text omitted]